# Xiphoid nucleus of the midline thalamus controls cold-induced food seeking

Neeraj K. Lal[1], Phuong Le[2], Samarth Aggarwal[1], Alan Zhang[1], Kristina Wang[1], Tianbo Qi[1], Zhengyuan Pang[1], Dong Yang[1], Victoria Nudell[1], Gene W. Yeo[2], Alexander S. Banks[3] & Li Ye[1,4 ✉]

Maintaining body temperature is calorically expensive for endothermic animals[1]. Mammals eat more in the cold to compensate for energy expenditure[2], but the neural mechanism underlying this coupling is not well understood. Through behavioural and metabolic analyses, we found that mice dynamically switch between energy-conservation and food-seeking states in the cold, the latter of which are primarily driven by energy expenditure rather than the sensation of cold. To identify the neural mechanisms underlying cold-induced food seeking, we used whole-brain c-Fos mapping and found that the xiphoid (Xi), a small nucleus in the midline thalamus, was selectively activated by prolonged cold associated with elevated energy expenditure but not with acute cold exposure. In vivo calcium imaging showed that Xi activity correlates with food-seeking episodes under cold conditions. Using activity-dependent viral strategies, we found that optogenetic and chemogenetic stimulation of cold-activated Xi neurons selectively recapitulated food seeking under cold conditions whereas their inhibition suppressed it. Mechanistically, Xi encodes a context-dependent valence switch that promotes food-seeking behaviours under cold but not warm conditions. Furthermore, these behaviours are mediated by a Xi-to-nucleus accumbens projection. Our results establish Xi as a key region in the control of cold-induced feeding, which is an important mechanism in the maintenance of energy homeostasis in endothermic animals.

The emergence of endothermy brought numerous adaptive advantages during evolution; however, it also came with a significant increase in energy expenditure. To fuel this increased energy demand, mammals dramatically adapt their foraging behaviour in response to changing temperature and there is a tight, inextricable association between ambient temperature and food intake: the colder the environment, the more food is needed to maintain core body temperature[1–3]. Mammals, including humans, are known to eat more in the cold. Various festivals across different cultures involving lavish feasts during winter are a testimony to our endothermic evolutionary past[4–6]. However, the neural basis linking the energy needs arising from the cold and the increase in feeding remain unanswered questions in our understanding of mammalian biology.

Rodents are an excellent model with which to study this association between temperature and energy consumption[7]. For example, laboratory mice (*Mus musculus*) can double their daily food intake when living at 4 °C, with thermogenesis contributing to around 60% of whole-body energy expenditure under these conditions[8]. Although cold sensation has been shown to acutely influence feeding through conserved somatosensory and feeding centres in the brain in both ectotherms and endotherms[9–13], it remains unclear whether—or how—cold-induced energy expenditure is compensated by food intake. Here we combined high-resolution metabolic and behavioural analyses to demonstrate that increased feeding in the cold is a consequence of

energy expenditure; moreover, we identified the midline thalamic xiphoid nucleus (Xi) as a key hub mediating the compensatory increase in food-seeking behaviours.

## Energy expenditure drives feeding in cold conditions

Housing mice at 4 °C has been reported to lead to elevated thermogenesis, energy expenditure and food intake[8,14]. To gain a more detailed view of energy expenditure, we used indirect calorimetry to determine an individual mouse's energy expenditure in real time through the measurement of oxygen and carbon dioxide exchanges[15]. Temporally resolved indirect calorimetry showed that switching the environment temperature from 23 to 4 °C immediately increased energy expenditure (Fig. 1a,b and Extended Data Fig. 1a–f). However, there was an initial decrease and substantial delay between the temperature drop and increased food intake (Fig. 1b), suggesting that rapid cold sensation might not be the direct cause of cold-induced feeding. Following quantification of the temporal association between energy expenditure and feeding we found that, as cold exposure progressed, food intake became more correlated with energy expenditure (starting at 5–6 h post cold exposure and remaining elevated thereafter; Fig.1c). On the basis of this progressive correlation, we thus defined this 5–6 h post-cold-exposure period as the onset of cold-induced energy

[1]Department of Neuroscience, The Scripps Research Institute, La Jolla, CA, USA. [2]Department of Cellular and Molecular Medicine, University of California San Diego, La Jolla, CA, USA. [3]Division of Endocrinology, Diabetes and Metabolism, Beth Israel Deaconess Medical Center, Boston, MA, USA. [4]Department of Molecular Medicine, The Scripps Research Institute, La Jolla, CA, USA. ✉e-mail: liye@scripps.edu

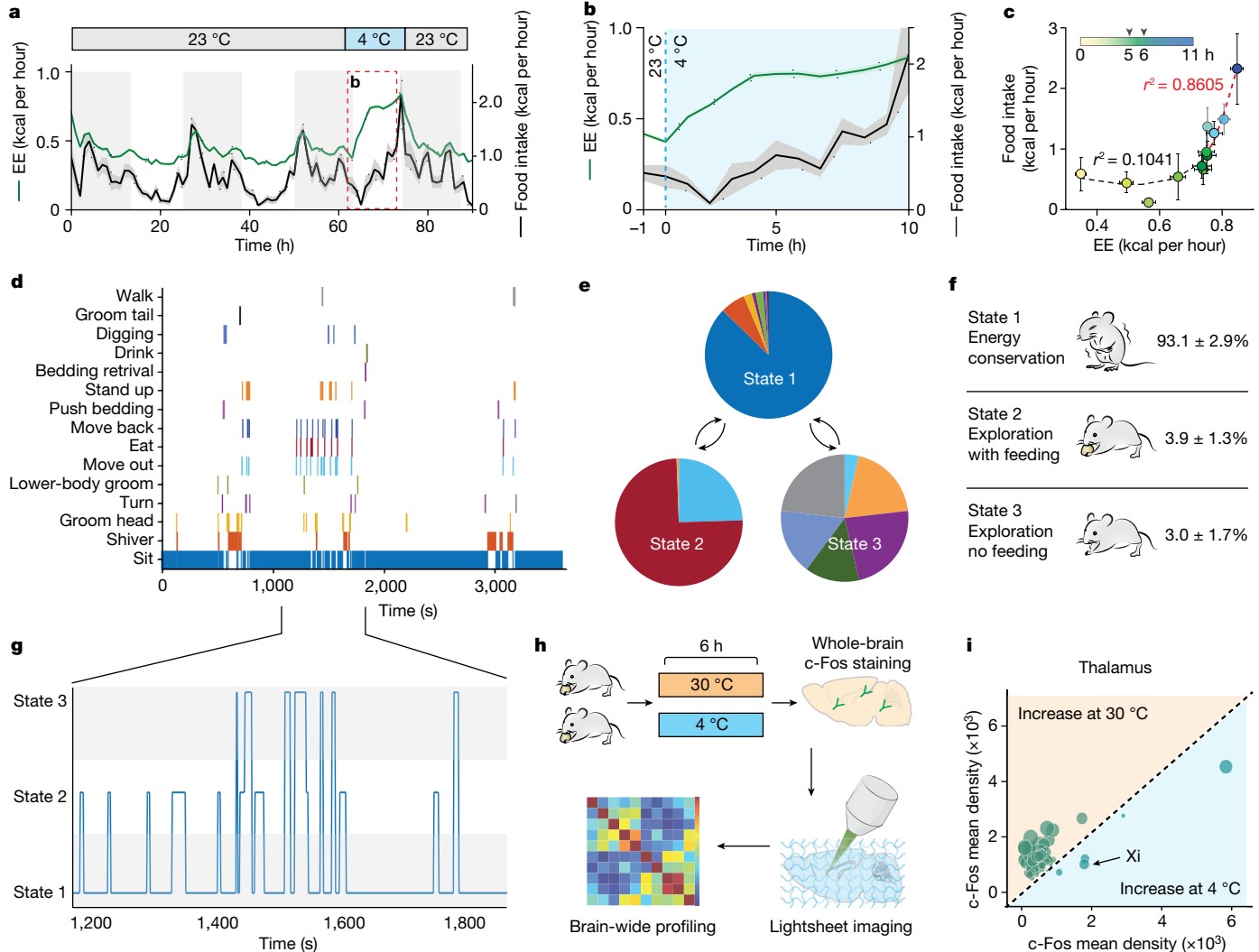

**Fig. 1 | Identification of the behavioural and circuit basis of cold-induced feeding. a**, Ninety-hour progression of energy expenditure (EE) and food intake (*n* = 24 mice). Energy expenditure is represented in green and food intake in black; lines and shading denote mean and s.e.m., respectively, of each group. **b**, Enlarged view of **a** during the switch from 23 to 4 °C. **c**, Scatter plot representing the relationship between energy expenditure and feeding over an 11 h period post temperature switch (during the light cycle, *n* = 24 mice). Each dot represents average energy expenditure and food intake across all mice at each hour. Arrowheads indicate the key transition period (at 5–6 h) after onset of cold conditions. **d**–**f**, Behavioural analysis of animals undergoing CIEC using a HMM. We defined a three-state HMM: (1) energy conservion; (2) exploration with feeding; and (3) exploration without feeding. We generated the initial estimated transition matrix by assuming equal probabilities for all transitions because we did not have a priori information. **d**, Representative behavioural events assessed for a male C57BL/6J mouse after being under cold conditions for 5 h. **e**, Representation of the HMM of CIEC-associated feeding (*n* = 3 mice). **f**, Percentage of time spent in each HMM state. **g**, Enlarged view from a 10 min session of **e**, showing HMM state assignment. **h**, Schematics of whole-brain clearing and volumetric three-dimensional imaging used to identify brain regions activated during CIEC. **i**, c-Fos mapping results for the thalamus. Each dot represents c-Fos+ cell count in each distinct region based on Allen Brain Atlas registration; dot size represents the difference in signal density between the two conditions. Structures activated under thermoneutral temperature are shaded in orange and those activated under cold conditions in blue.

compensation (CIEC) and focused on this time window for the remainder of the study, including video recording of mice within this period to better understand these behaviours.

Interestingly, despite the overall increase in food intake when experiencing cold, the mice mostly stayed immobile rather than actively engaging in foraging (Fig. 1d). This observation suggested that, rather than a unilateral increase in appetite, the mice faced competing priorities between conserving energy for thermogenesis (by staying immobile) versus replenishing energy supply (by seeking food). However, mice also engaged in various non-feeding, thermoregulatory behaviours under cold conditions, such as bedding retrieval and arrangement (Fig. 1d). To better understand the relationships among these behaviours, we first annotated them into 15 specific actions and used an unsupervised hidden Markov model (HMM) to identify three distinct states shown

by the mice: state 1, an energy-conservation state in which they were mostly immobile; state 2, an exploration with food-seeking state with the highest probability of eating; and state 3, an exploration without food-seeking state, where mice engaged in other actions without food consumption (Fig. 1d–g and Extended Data Fig. 2a–c). This analysis enabled us to use just three HMM states, rather than the 15 specific actions, to selectively study CIEC-induced feeding state (state 2). Because state 1 is the predominant state (Fig. 1f), we focused on characterization of outbound state transitions from state 1 for all subsequent analyses.

## Brain-wide activity screen under cold conditions

To identify the circuit mechanisms underlying CIEC-induced feeding, we first performed a brain-wide c-Fos screen[16,17] on mice that had been

kept at either 4 or 30 °C (murine thermoneutrality, for maximization of brain-wide activity differences) for 6 h, using whole-brain SHIELD (stabilization under harsh conditions via intramolecular epoxide linkages to prevent degradation), immunofluorescence labelling, lightsheet imaging and automated analysis (Fig. 1h and Supplementary Table 1)[18–20]. Cold conditions led to a broad decrease in c-Fos signal in the cortex, probably due to the overall decrease in physical activity of these animals (Extended Data Fig. 3a–c). The hypothalamus, where thermoregulation is centred[21], showed robust c-Fos activation as expected (Extended Data Fig. 3a,d). Surprisingly, whereas most of the thalamus was suppressed under cold conditions, several ventral midline thalamic (vMT) regions showed higher activation (Fig. 1i). We decided to further explore this area for two reasons. First, in the Siberian hamster (a model for cold adaptations) lesions at the vMT have been reported to impair cold adaptation in winter[22]. Second, it was recently shown in mice that the vMT is a crucial hub for gating how internal states are translated into opposite behaviour responses (for example, freezing versus fighting) to perceived threats[21], a scenario that resembles the competing priorities between energy-conservation and food-seeking behaviours we observed in CIEC.

Next, to differentiate whether vMT activation occurred because of cold sensation or was related to CIEC, we examined c-Fos expression after either 5 h (to model CIEC) or 15 min (to model acute sensation) of cold exposure (Fig. 2a). Both conditions induced c-Fos in the median preoptic nucleus, a key region for thermoregulation (Extended Data Fig. 4b). However, as little as 5 h of cold exposure led to c-Fos activation in the Xi compared with 15 min cold exposure or housing at 30 °C; this activation was not observed in the neighbouring nucleus reuniens (Fig. 2b,c and Extended Data Fig. 4a). Following our designation of the 5–6 h post-cold-exposure period as the onset of CIEC (Fig. 1c), we define these c-Fos⁺ Xi neurons as the Xi^CIEC population, whose activation is probably due to CIEC rather than to acute cold sensation. This selective Xi activation was also observed after an extended period of cold (7 days) (Extended Data Fig. 4c). Furthermore, both male and female mice showed activation of the Xi during CIEC (Extended Data Fig. 4d).

## Xi activity during CIEC-induced feeding

To determine how Xi activity represents different behavioural components induced by cold conditions in real time, we used fibre photometry to record in vivo calcium dynamics of the Xi (Fig. 2d,e). Consistent with the c-Fos results (Fig. 2b,c), we found that an acute decrease in temperature (23 to 4 °C) did not lead to activation of the Xi (Extended Data Fig. 4e). Next, we analysed Xi calcium dynamics during CIEC-associated food seeking. Freely moving mice were placed under cold conditions with food and water for 5 h before fibre photometry recording. There was a marked increase in Xi activity before each feeding event under cold conditions, but this was not observed in feeding bouts at thermoneutral temperature (non-CIEC) in the same individuals (Fig. 2f–i), suggesting that Xi activity was selectively associated with CIEC-induced feeding. To gain additional insight into cold-induced Xi activity, we created a scenario with exacerbated cold-induced energy deficit (CIEC⁺) by briefly restricting food access during the first 3 h of cold exposure. Under CIEC⁺ conditions we observed further elevated Xi calcium transients within the same animals, suggesting that Xi activity was scalable with CIEC (Extended Data Fig. 4f–h). Furthermore, Xi neurons did not respond to canonical, fasting-induced feeding at either room temperature or 30 °C (Extended Data Fig. 4j–l), suggesting that their activity is specifically associated with CIEC but not with a general energy deficit associated with food restriction. Plasma levels of glucose and leptin, which normally decrease with fasting, were unaltered during CIEC (Extended Data Figs. 1j and 5a,b) whereas there were elevated levels of lipolysis products (Extended Data Fig. 5c–f), suggesting that non-canonical pathways are signalling the CIEC state

to the brain, although the exact molecular nature of such a pathway remains to be determined.

By further incorporation of HMM states into fibre photometry analysis, we discovered that Xi activity is strongly associated with transition between the energy-conserving state (state 1) and food-seeking state (state 2), but not with transition between state 1 and the non-food-seeking, exploratory state 3 (Fig. 2j–l and Extended Data Fig. 4i), suggesting that Xi activity is not associated with general exploratory movements. Together these results indicate that Xi neurons are specifically activated during the transition between states 1 and 2, leading to compensatory feeding in a manner that scales with increasing amplitude at higher CIEC.

## Xi regulates CIEC-induced feeding

The temporal relationship between Xi activity and CIEC-induced food seeking led us to ask whether Xi neurons could causally modulate this behaviour. However, the Xi is a small, less-studied midline thalamic nucleus without well-defined boundaries or molecular markers[23,24]. Noting that CIEC selectively induced c-Fos in the Xi without activation of the surrounding areas (Fig. 2b), we hypothesized that we could target these Xi^CIEC neurons based on their unique activation history during previous exposures to cold conditions. Leveraging a previously validated vCAP-TURE strategy[17,25–28] based on activity-dependent ESARE-ER-Cre-ER[29], we targeted cold-activated Xi neurons with an hM3Dq (Gq-coupled human muscarinic M3 designer receptor exclusively activated by either a designer drug (DREADD) or a red fluorescent protein (RFP) control; Fig. 3a–c). Consistent with the specificity and efficiency of previous CAPTURE and targeted recombination in active populations work[25,27,29], we found that Xi^CIEC neurons were efficiently captured at 4 °C and that these neurons highly overlapped with cold-induced c-Fos⁺ neurons (89.9 ± 6.7% by c-Fos/CAPTURE, 52.6 ± 12.4% by CAPTURE/c-Fos), whereas minimal neurons were captured at 30 °C (Figs. 3a,d–f and 2a,b).

Following reactivation by clozapine N-oxide (CNO), we observed significantly increased food intake in mice targeted with hM3Dq under cold conditions (Xi^CIEC) compared with those targeted with the RFP control or at 30 °C (Xi^non-CIEC) (Fig. 3g). We then used the same vCAPTURE strategy to target Xi^CIEC neurons with an inhibitory hM4Di DREADD. A significant decrease in CIEC-induced food intake was observed following CNO-mediated inhibition compared with the RFP control (Fig. 3h–j and Extended Data Fig. 6a). By contrast, neither chemogenetic inhibition nor activation affected food intake at room temperature (Extended Data Fig. 6b,c), further indicating that the role of Xi neurons is specific to cold-induced feeding.

## Xi^CIEC neurons encode a context-specific valence

To understand how Xi^CIEC neurons promote feeding at higher temporal resolution, we turned to optogenetics. We first injected ChR2-expressing AAV directly into the vMT region and analysed feeding behaviours in a CIEC scenario (Extended Data Fig. 7a,b). Optogenetic stimulation of the bulk vMT region led to higher food intake but also caused a rapid increase in overall movement that was not observed during natural adaption to cold conditions (Extended Data Fig. 7c–f). These results suggested that non-specific activation of neighbouring neurons outside the Xi may have confounded behavioural output.

To improve specificity we again used the activity-dependent vCAP-TURE system to selectively express ChR2 in Xi^CIEC neurons (Fig. 4a,b). Photostimulation of ChR2-expressing Xi^CIEC neurons resulted in a robust increase in food intake without changes in general movement (Fig. 4c–f) or in an open-field test (Extended Data Fig. 8a–d) compared with RFP controls. Interestingly, in the absence of cold (that is, at rodent thermoneutrality, 30 °C), photostimulation of Xi^CIEC neurons resulted in only a minor, but non-significant, change in food intake (Extended Data Fig. 8e–h), indicating that Xi^CIEC neurons require a CIEC

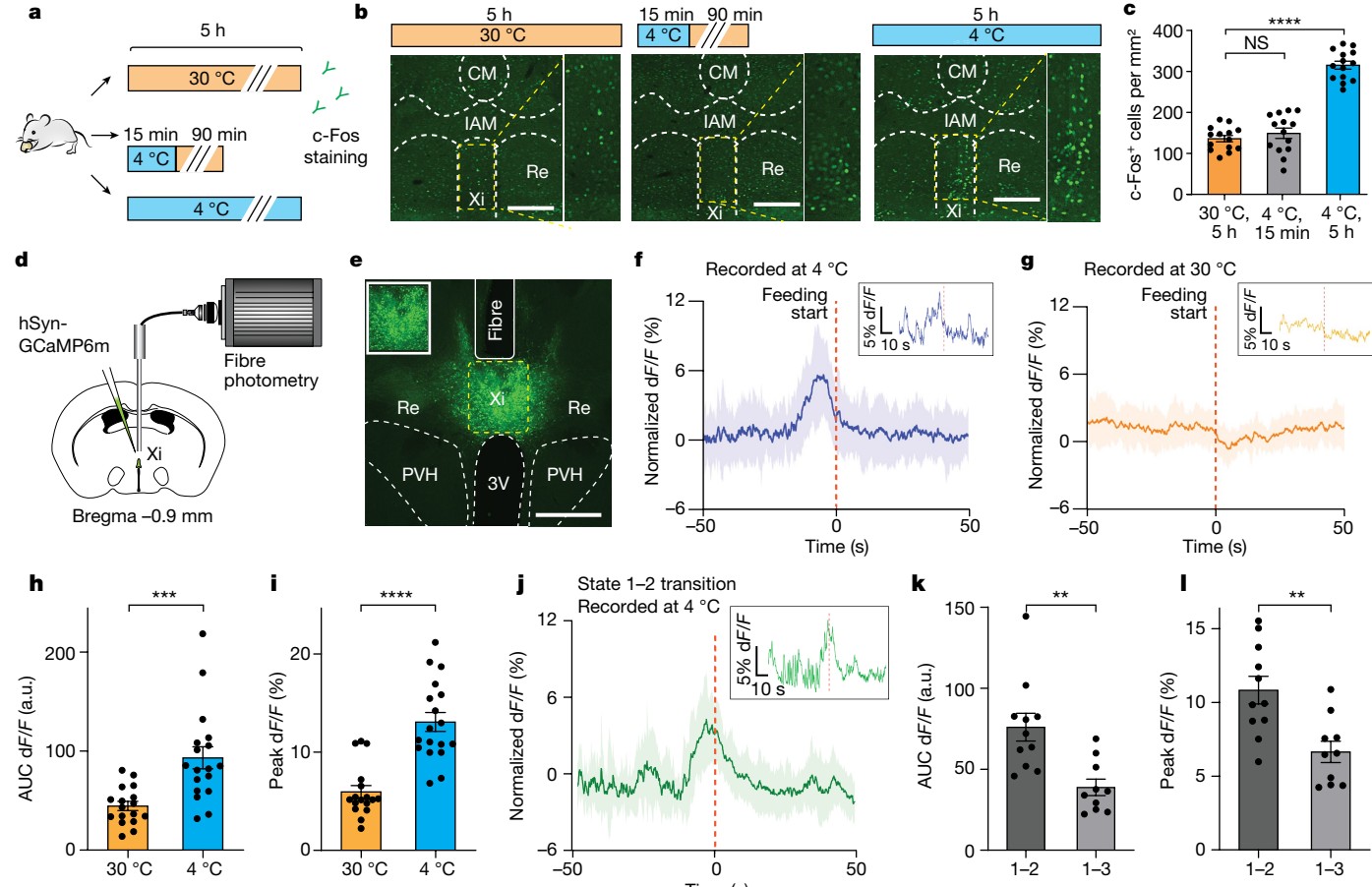

**Fig. 2 | Xi activation is associated with CIEC-induced feeding. a,** Schematic showing different cold-exposure paradigms used to label c-Fos expression. **b,** c-Fos⁺ neurons (green) in the vMT for each condition shown in **a.** Yellow-bordered boxes show enlarged images of the Xi region. Scale bars, 200 μm. CM, central medial nucleus; IAM, intrantereomedial nucleus; Re, nucleus reuniens. **c,** Quantification of c-Fos⁺ cells under each condition (*n* = 14 sections from four mice for each condition). Data are mean ± s.e.m. ****P < 0.0001; NS, non-significant using ordinary one-way analysis of variance (ANOVA) with Dunnett's multiple comparison test. **d,** Schematic of fibre photometry setup for recording from the Xi. **e,** Representative image of GCaMP6m-labelled Xi neurons. Scale bar, 500 μm. **f,g,** Fibre photometry signal of AAV-GCaMP6m-expressing vMT/Xi neurons under cold (**f**) and thermoneutral (**g**) conditions,

shown as the average of 18 events from four different mice. Solid line represents the average, and shaded area the s.e.m.; red dashed line represents the start of feeding. Insets show an example of a single trace. **h,i,** Bar graphs of the area under the curve (AUC) d*F*/*F* (−20 to 10 s) (**h**) and peak d*F*/*F* percentage (**i**) for fibre photometry data in **f** and **g**, respectively. **j,** Fibre photometry signal aligned to state 1–2 transition (red dashed line) based on HMM, averaged from 11 events from three different mice; solid line represents average and shaded area is s.e.m. Inset shows an example of a single trace. **k,l,** Bar graphs of AUC (−10 to 10 s) (**k**) and peak d*F*/*F* (**l**) quantified from **j** and Extended Data Fig. 4. Data are mean ± s.e.m. ***P = 0.004 for AUC (**h**), ****P < 0.0001 for peak d*F*/*F* (**i**), **P = 0.0017 for AUC (**k**) and **P = 0.0024 for peak d*F*/*F* (**l**) using two-tailed unpaired *t*-test. a.u., arbitrary units.

state to induce bona fide feeding in animals. In addition, video analysis showed that photoactivation of Xi^CIEC resulted in more transitions from the energy-conserving HMM state 1 to the food-seeking state 2, whereas state 1–3 transitions were not altered (Fig. 4g,h). Similarly, the total time spent in state 2 increased following photostimulation (Fig. 4i and Extended Data Fig. 8i). These results, in combination with the earlier fibre photometry data, indicate that Xi^CIEC activity specifically promotes CIEC-induced feeding without affecting other general movement-related actions.

Next, to delineate how Xi neurons affect behavioural transitions, we used real-time place preference (RTPP) to determine the valence associated with Xi activation. Xi^CIEC ChR2-expressing mice were first placed in a two-chamber arena without laser stimulation to determine a basal place preference. During RTPP, Xi^CIEC neurons were stimulated when the mice entered one side of the arena and this stopped when they crossed to the other side (Fig. 4j,k). At room temperature, Xi^CIEC activation did not change place preference. However, when RTPP was conducted under cold conditions (4 °C), the same animals switched

their preference to the light-stimulated side (Fig. 4j–m). This result is consistent with our state-dependent activity pattern found in earlier fibre photometry studies (Fig. 2f,g). Moreover, such a state-dependent valence switch was present only in ChR2-expressing mice and not in RFP controls. The RTPP results suggest that Xi-mediated state transition for food seeking during CIEC could be driven by a positive valence.

## Xi-NAc mediates food seeking under cold conditions

Lastly, we sought to characterize the cell types and projection targets of the Xi. First, using vGLUT2-Cre and vGAT-Cre mice[30,31], we found that Xi-regulated food seeking was primarily mediated by glutamatergic neurons in the Xi and not through GABAergic cells (Extended data Fig. 9a–l). Next we injected mCherry-expressing adeno-associated virus (AAV) to map the projection targets of Xi. Consistent with previous reports[23,24], Xi projected to the nucleus accumbens (NAc), baso-lateral amygdala (BLA) and anterior cingulate cortex (ACC) (Fig. 5a,b). However, we should note here that because Xi is a small region and

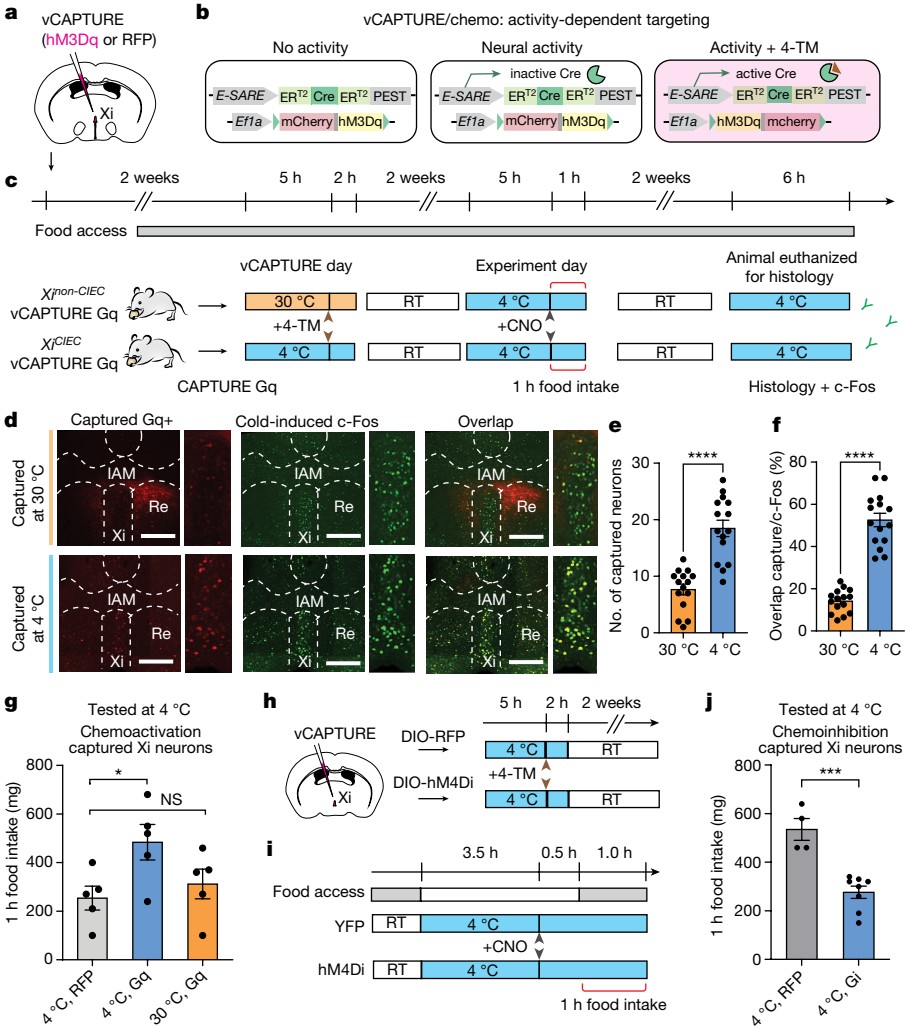

**Fig. 3 | Xi neurons modulate CIEC-induced feeding. a**, Schematic showing injection site for AAV-eSARE-ER-Cre-ER and AAV-DIO-hM3Dq or RFP control in the Xi. **b**, Schematics of vCAPTURE activity-dependent labelling strategy used to express hM3Dq DREADD in CIEC-activated Xi neurons. **c**, Schematics showing procedures for activity-dependent capture of CIEC versus non-CIEC neurons in the Xi. For vCAPTURE of Xi^CIEC neurons, tamoxifen was given to mice 5 h after exposure to cold conditions (4 °C). RT, room temperature. **d**, Representative histology from non-CIEC (30 °C, top) and CIEC (4 °C, bottom) mice used for c-Fos double labelling. Left: vCAPTURE-mediated hM3Dq labelling in the vMT/ Xi region; middle: CIEC-activated c-Fos labelling; right: overlay of vCAPTURE and c-Fos double labelling. Scale bars, 500 μm **e,f**, Quantification of c-Fos double labelling with vCAPTURE neurons in the Xi (16 sections from $n$ = 4 mice). Captured-Gq total numbers (**e**) and overlap with 4 °C c-Fos (**f**). Data are

mean ± s.e.m. ****$P$ < 0.0001 using two-tailed unpaired $t$-test. **g**, Bar graph showing food-intake levels following activation of Xi^CIEC versus Xi^non-CIEC neurons compared with RFP controls ($n$ = 5 mice per group). Data are mean ± s.e.m. *$P$ = 0.0397 and NS = 0.7374 using ordinary one-way ANOVA with Dunnett's correction for multiple comparisons. **h,i**, Schematics of vCAPTURE strategy (**h**) and procedure (**i**) used to test loss-of-function experiment with DREADD-Gi(hM4Di). Note that mice in this experiment were briefly food restricted (indicated by grey bars) before injection of CNO, to elevate baseline feeding. YFP, yellow fluorescent protein. **j**, Bar graph showing food-intake levels following inhibition of Xi^CIEC neurons ($n$ = 4 mice for RFP and $n$ = 4 for Gi). Data are mean ± s.e.m. ***$P$ = 0.0003 using two-tailed unpaired $t$-test. 4-TM, 4-hydroxytamoxifen.

we currently do not have a genetic marker for CIEC-activated Xi neurons, our tracing studies have a limitation in regard to targeting of neighbouring regions such as PVH and Re. To further confirm and determine whether one or more of these projections corresponded to the Xi^CIEC population, we designed a double-labelling experiment in which retrograde CTB dyes were individually injected into the NAc, BLA and ACC. We found that the highest overlap between cold-activated c-Fos and CTB was in Xi-NAc projecting neurons (Fig. 5d–g), prompting us to test the role of Xi-NAc projection in cold-induced feeding. We injected ChR2-expressing AAV into the Xi and implanted an optic fibre above either the NAc, BLA or ACC (Fig. 5i,j, top and Extended Data Fig. 9a). Photoactivation of Xi-NAc projection, but not Xi to ACC or BLA projection, resulted in a significant increase in food intake (Fig. 5i,j and Extended Data Fig. 10a,b). These projection-specific effects were

further confirmed using two fibre implants (NAc and BLA) in the same animals but photostimulated sequentially (Extended Data Fig. 10c–g). Finally, using RTPP, we also found that activation of Xi to NAc projection resulted in a cold-dependent positive valence whereas activation of other projections failed to do so (Fig. 5k,l). Together, these results show that Xi-to-NAc projection primarily mediates cold-induced food-seeking behaviours.

## Discussion

Through leverage of whole-brain screening, in vivo calcium imaging and chemo- and optogenetic manipulations, we demonstrated that the Xi nucleus serves as a key brain region in the promotion of cold-induced food-seeking behaviours. Although cold-induced feeding has been

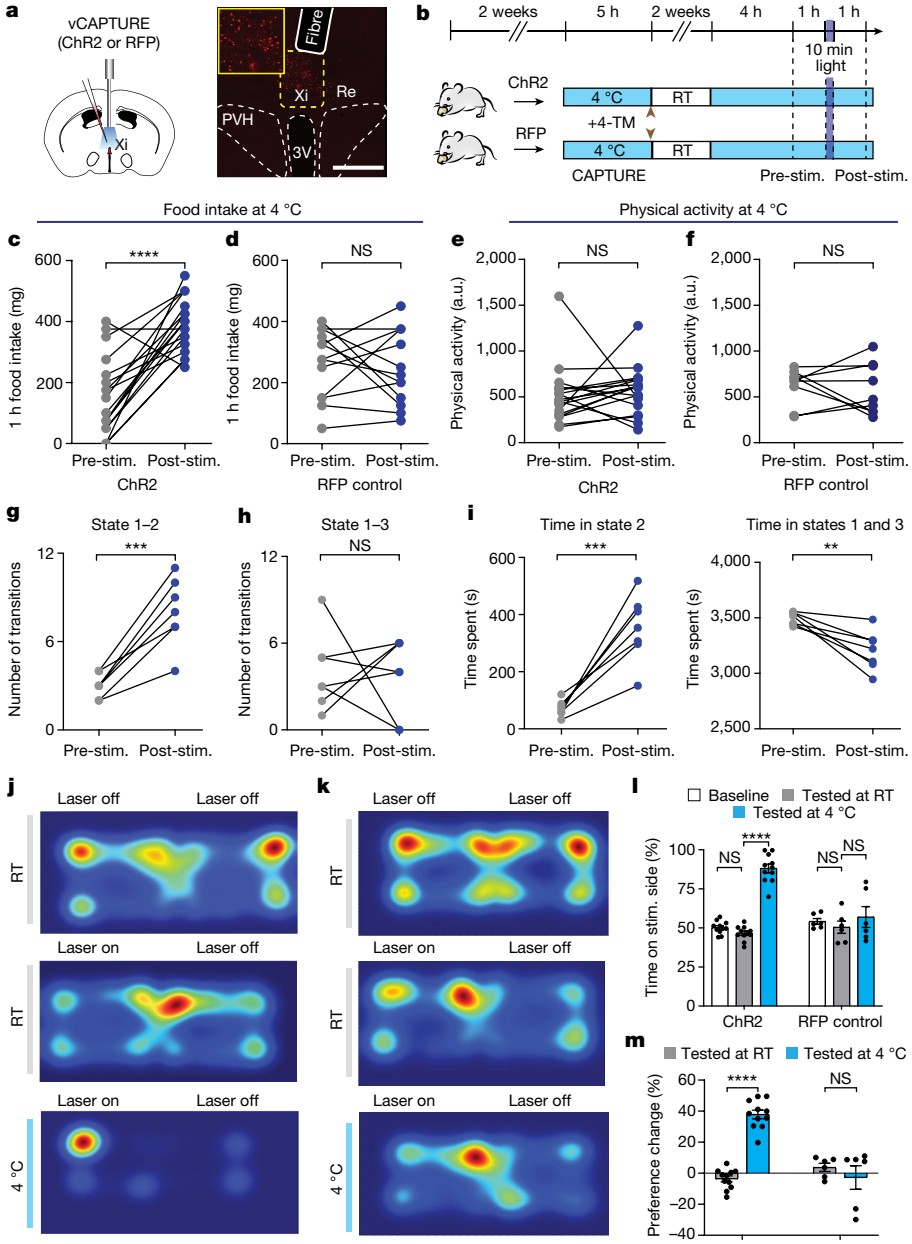

**Fig. 4 | Xi^CIEC activity represents a context-specific valence. a**, Schematic of viral injection, fibre placement, and representative histology images. Scale bar, 500 μm. **b**, Schematic showing vCAPTURE optogenetic experimental procedures. Stim., stimulation. **c,d**, Difference in food intake under cold conditions for ChR2 mice (*n* = 9 mice, *n* = 18 times) (**c**) and for RFP control mice (*n* = 7 mice, *n* = 13 times) (**d**). **e,f**, Difference in physical activity under cold conditions for ChR2 mice (*n* = 9 mice, *n* = 18 times) (**e**) and for RFP control mice (*n* = 7 mice, *n* = 9 times) (**f**). Data are mean ± s.e.m. ****$P$ < 0.0001 (**c**), NS = 0.6953 (**d**), NS = 0.6287 (**e**) and NS = 0.7252 (**f**) using two-tailed paired *t*-test. **g**, Total numbers of transitions from state 1 (energy-saving state) to state 2 (exploration with feeding) in ChR2 mice (*n* = 7 mice). **h**, State transition from state 1 to state 3 (exploration without feeding). **i**, Total time spent in different states (*n* = 7 mice). Data are mean ± s.e.m. ***$P$ = 0.0005 (**g**), NS = 0.8779 (**h**), ***$P$ = 0.0006

(**i**, left), and **$P$ = 0.0046 (**i**, right) using two-tailed paired *t*-test. **j**–**m**, RTPP test. **j,k**, Representative heatmaps of a ChR2 mouse (**j**) and RFP control (**k**). **l**, Quantification of percentage of time spent on the stimulated side. Data are mean ± s.e.m. ****$P$ < 0.0001 for ChR2 baseline versus ChR2 4 °C and ChR2 RT versus ChR2 4 °C; NS = 0.4589 for ChR2 baseline versus ChR2 RT; NS = 0.7083, 0.9985 and 0.9648 for RFP baseline versus RT, 4 °C and RFP RT versus 4 °C, respectively, using ordinary two-way ANOVA with Tukey's multiple comparison test (*n* = 11 for ChR2 and *n* = 6 for RFP control). **m**, Percentage change in preference between time spent in stimulation chamber, normalized to laser-off basal level (*n* = 11 for ChR2 and *n* = 6 for RFP control). Data are mean ± s.e.m. ****$P$ < 0.0001 for ChR2 RT versus ChR2 4 °C, NS = 0.6887 for RFP RT versus RFP 4 °C using ordinary two-way ANOVA with Tukey's multiple comparison test.

widely reported across animal species, including humans, our study suggested that such a feeding increase was neither a direct response to cold sensation nor a unilateral increase in food-seeking behaviours but rather a dynamic outcome of the transitional priorities between energy conservation and replenishment in response to elevated caloric debt in cold conditions.

Interestingly, although Xi activity was associated with behavioural transitions (Fig. 2j), it returned to baseline before feeding began (Fig. 2f), suggesting that sustained Xi activity is not required for feeding. The vMT/Xi has recently been shown to gate behavioural shifts between saliency-reducing and -enhancing strategies in response to visual threats in mice[25]. Reminiscent of this shift, we speculated that

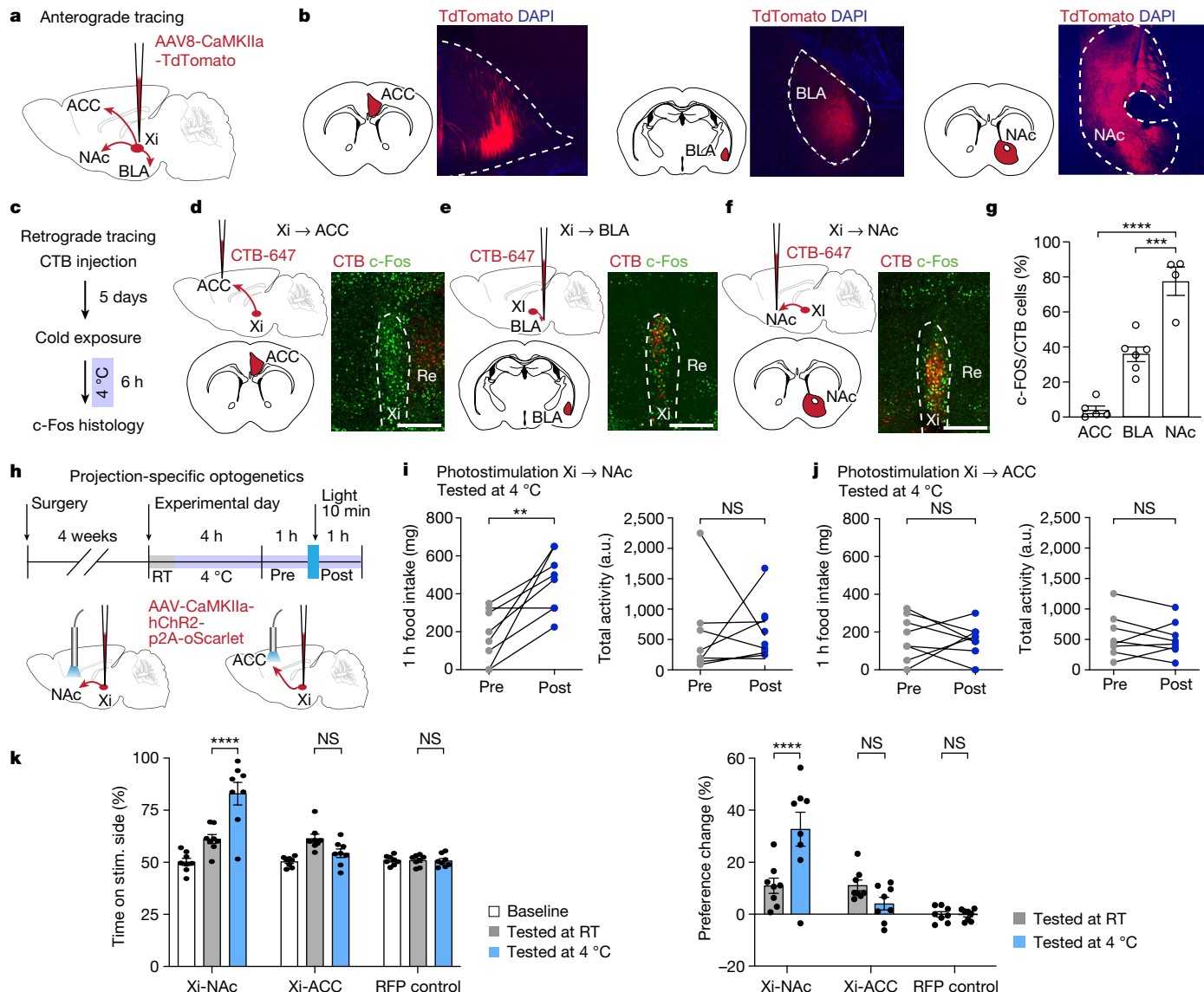

**Fig. 5 | Xi-to-NAc projection mediates CIEC-associated food seeking.**
**a**,**b**, Schematic for anterograde tdTomato labelling (**a**). Mice were injected with AAV-CaMKIIa-tdTomato at the Xi and axonal fibres observed in the ACC, BLA and NAc (**b**) (*n* = 5 mice). **c**–**f**, Schematic (**c**) and representative overlay images between retrograde CTB and c-Fos signals at the Xi: ACC-Xi (**d**), BLA-Xi (**e**) and Xi-NAc (**f**). Scale bars, 200 μm. **g**, Quantification of CTB/c-Fos overlap. *n* = 5 for ACC, *n* = 6 for BLA and *n* = 4 for NAc. Data are mean ± s.e.m. ****P < 0.0001 for ACC versus NAc and ***P = 0.0002 for BLA versus NAc using ordinary one-way ANOVA with Tukey's multiple comparison test. **h**. Schematic of projection-specific optogenetics experiments. **i**,**j**, Quantification of change in food intake and physical activity following stimulation of Xi-NAc (**i**) and Xi-ACC (**j**) projection during CIEC (*n* = 8 mice). Data are mean ± s.e.m. **P = 0.0052 for food intake,

NS = 0.9163 for physical activity for Xi-NAc (**i**); NS = 0.8037 for food intake and NS = 0.5893 for physical activity for Xi-ACC using a two-tailed paired *t*-test (**j**). **k**,**l**, Projection-specific RTPP test. RFP control was injected in the Xi and stimulated at the Xi to control for all projections (*n* = 8 mice). **k**, Percentage of time spent on laser stimulation side. Data are mean ± s.e.m. ****P < 0.0001 for Xi-NAc RT versus Xi-NAc 4 °C, NS = 0.3027 for Xi-ACC RT versus Xi-ACC 4 °C, NS > 0.99 for RFP RT versus RFP 4 °C (*n* = 8 mice per group). **l**, Percentage change in preference following stimulation, normalized to baseline place preference for the same animal. Data are mean ± s.e.m. ***P = 0.0003 for Xi-NAc RT versus Xi-NAc 4 °C, NS = 0.6413 for Xi-ACC versus Xi-ACC 4 °C and NS > 0.99 for RFP RT versus RFP 4 °C using ordinary two-way ANOVA with Tukey's multiple comparison test (*n* = 8 mice per group).

endogenous Xi activity might gate behavioural transitions from energy conservation to a replenishing (feeding) state, because experimental activation of Xi increased CIEC-associated feeding (Figs. 3g and 4c), potentially by lowering the threshold required to increase the probability of state 1–2 transitions (Fig. 4g–i). This gating, however, is specific to the cold context in which coordinating energy expenditure and intake is critical for survival: it appears to be less involved in starvation conditions, in which the drive for feeding is more unilateral (Extended Data Fig. 4). This specificity suggests that additional CIEC-associated inputs, such as metabolites, hormones or interoceptive signals, acting on one or more upstream regions of Xi, are necessary for Xi-mediated

behavioural gating and valence transitions. Further studies are required to elucidate the identities and mechanisms of these inputs. In concert with all available evidence, we propose that the Xi could represent an important mechanism for dynamic switching of opposing survival strategies during specific natural behaviours.

Feeding behaviours are tightly regulated by both homeostatic and reward circuits[32–34]. The Xi is a less-studied brain region that is not a direct component of the classic hypothalamic or limbic reward systems. However, it projects to multiple well-known regions involved in feeding regulation, including the prefrontal cortex, BLA and NAc[21] (Fig. 5a,b). We found that Xi-regulated feeding and valence-switching behaviours

were primarily mediated by the Xi to NAc projection. As a limbic reward centre, the NAc is known to integrate motivational and sensory signals to guide motor output[35,36] and regulate feeding[37,38]. On the basis of our findings, we propose that the NAc is a key downstream target of the Xi in motivation of animals to switch behavioural states as dictated by the internal metabolic state. It remains to be determined, however, how Xi activity interacts or integrates with other aspects of the canonical feeding framework and, especially, what the identity of its input signals are.

Although feeding is one of the most extensively studied rodent behaviours, it has primarily been modelled in the context of food restriction (for example, overnight fasting) in the creation of negative energy balance. Cold-induced feeding provides a new paradigm to study appetite that is driven by energy expenditure. Although both food restriction and energy expenditure create a net negative energy balance, the organismal physiology can be markedly different—for example, fasting is associated with lower blood glucose, leptin and insulin and suppressed thermogenesis and basic metabolic rates, whereas cold adaptation is accompanied by high glucose utilization, elevated thermogenesis and higher metabolic rates. Cold-induced energy compensation and associated neural substrates (such as the Xi and many others shown by the whole-brain screen) could provide an entry point in understanding naturalistic energy deficit created by other high-EE states such as exercise and lactation, findings that will be important not only for understanding fundamental mammalian biology but also for obesity and weight management.

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

## Methods

### Mice

Animal experiments were performed either at Scripps Research Institute, La Jolla or Beth Israel Deaconess Medical Center. Experiments were approved by either the Scripps Research Institute's or Beth Israel Deaconess Medical Center's Institutional Animal Care and Use Committee, respectively. Experiments were performed on mice aged 2–6 months. Wild-type C57BL/J6 mice were injected with viruses between at 6–8 weeks of age and were allowed to recover for at least 1 week before experiments or experiment-related habituation. Group size was not predetermined based on statistical power law but on previously published studies. Animals were randomly assigned to different experimental groups before viral injections. Both males and females were used for histology and c-Fos labelling. Male mice were used for free-moving behavioural studies.

### Reagents and viruses

AAV8-hSyn-DIO-Gq–mCherry (Addgene, no.44361-AAV8), AAV8-hSyn-DIO-Gi–mCherry (Addgene, no.44362-AAV8), AAV8-hSyn-mCherry (Addgene, no. 114472-AAV8) and AAV9-Syn-GCaMP6m-WPRE-S V40 (Addgene, no. 100841-AAV9) were obtained from Addgene.

AAV5-ESARE-ER-Cre-ER-PEST was obtained from UNC, and AAV8-Ef1a-DIO hChR2(H134R)-p2AScarlett and AAV8-CaMKIIa-hChR2-p2A-oScarlet were obtained from Stanford. c-Fos antibody was purchased commercially (Cell signaling, no. 2250) as was CNO (Hello, no. HB1807).

### Stereotactic viral injection and fibre implantation

Mice were anaesthetized with 3.0% isoflurane in an induction chamber and, after 5 min, the animals' heads were fixed on a stereotactic device (Kopf) using ear bars. Mice were maintained under 0.8–1.2% isoflurane during surgery on a warm heating pad (body temperature was maintained at 36 °C). Heads were shaved to remove hair and, using a scalpel blade, a midline incision was made to expose the skull. The head was balanced using a bregma and lambda system on the stereotaxic headstand. Using dental drills, small craniotomies were made above the site of injection. The virus was infused at 75 nl min$^{-1}$ using a Nanoject II (Drummond) injector connected to a 10 µl Hamilton syringe. The needle was retained at the injection site for 10 min before withdrawal. Mice were allowed to recover for 1–2 weeks before experiments or experiment-related habituation.

For chemogenetic stimulation, a cocktail of 125–150 nl of AAV8-hSyn-DIO-Gq–mCherry or AAV8-hSyn-mCherry and AAV5-ESARE-ER-Cre-ER-PEST was delivered to the vMT/Xi region (bregma, −1.0 mm; midline, 0 mm; dorsal surface, −4.4 mm). The final concentration of ESARE viruses in the cocktail was $7 \times 10^{12}$ genomic copies ml$^{-1}$. Two weeks after surgery, mice were habituated to tamoxifan injection (vCAPTURE labelling of Xi$^{CIEC}$-associated neurons).

For optogenetics experiments with non-selective activation of the whole vMT/Xi region (bregma, −1.0 mm; midline, 0 mm; dorsal surface, −4.4 mm), 150 nl of AAV8-CaMKIIa-hChR2-p2A-oScarlet ($5 \times 10^{12}$ genomic copies ml$^{-1}$) was injected in the vMT/Xi region. After removal of the needle, an optical fibre cannula (200 µm diameter, 0.22 Numerical Aperture (NA), RWD Life Science) was implanted at the vMT/XI 100–200 µm dorsal to the injection site (bregma, −1.0 mm; midline, 0 mm; dorsal surface, −4.2 to −4.3 mm). The optical cannula was cemented to the skull with dental cement. For the activity-dependent optogenetics labelling cohort, 200 nl of a cocktail of AAV8-Ef1a-DIO hChR2(H134R)-p2AScarlett + AAV5-ESARE-ER-Cre-ER-PEST ($7 \times 1,012$ genomic copies ml$^{-1}$) was injected in the vMT/Xi region.

For fibre photometry experiments, 150–180 nl of AAV9-Syn-GCaMP6m-WPRE-SV40 ($7 \times 10^{12}$ genomic copies ml$^{-1}$) virus was injected using stereotaxic surgery at the vMT/Xi region (bregma, −1.0 mm; midline, 0 mm; dorsal surface, −4.4 mm) and an optical fibre cannula (400 µm diameter, 0.5 NA, RWD Life Science) was implanted (bregma, −0.9 mm; midline, 0 mm; dorsal surface, −4.2 mm). For both optogenetics and fibre photometry, only a single-fibre optic cannula was implanted because the Xi is a midline structure.

### Cold and warm exposure

A temperature-controlled rodent incubator (no. RIS28SSD, Powers Scientific), maintained at 4 °C, was used for cold exposure. For thermoneutrality conditions a temperature-controlled warm room set to 30 °C was used. For all temperature-exposure experiments, all mice were individually housed with a small amount of bedding materials to ensure that each individual received the same temperature exposure. For cold exposure, animals were directly transferred to the 4 °C incubator in a lidless cage to facilitate temperature equilibrium. Mice were provided free access to chow food and water during most cold- and warm-exposure experiments, except in the CIEC$^+$ condition and during inhibitory chemogenetic experiments, in which they were food restricted for various amounts of time as indicated in individual figures.

### Optogenetic, chemogenetic and food-intake measurement

Mice were allowed to recover for 7 days after optogenetic implant surgery for the non-selective optogenetics experiment, or for 14 days after tamoxifen injection for the vCAPTURE cohort of mice. Optogenetic mice were trained and habituated to pellets (20 mg pellets, no. F0071, Bioserv) along with the pellet-dispensing system FED2 (ref. 39) in their home cage for 5 days, before being provided free access to food pellets for 4 days to avoid development of a hedonic value for food pellets. The mice were further habituated to the behavioural chambers for 5 days with food pellets and FED2 before the first experiment. All food-intake experiments were performed between 08:00 and 15:00 to avoid the natural circadian cycle influence on feeding behaviour.

For optogenetics experiments, experimental mice were connected to a blue-light laser (473 nm) with an optical fibre (200 µm diameter, 0.22 NA, RWD Life Science). Light was delivered in 10 ms pulses at 20 Hz, 3 s on/2 s off, for 10 min. Light power at the fibre tip was 10 mW. For chemogenetics experiments, a regular chow pellet was used for measurement of food intake by weight to increase throughput. The stock solution of CNO was prepared freshly by dissolving in DMSO and was then diluted in saline; mice were injected with 3 mg kg$^{-1}$ CNO for activation of the Gq cohort and 5 mg kg$^{-1}$ for the inhibitory Gi cohort. Control mice were given the same amount of CNO as the experimental cohort.

### Indirect calorimetry

Metabolic cage data were recorded for 24 individually housed mice placed in a Promethion indirect calorimeter (Sable Systems) with a temperature-controlled cabinet (Pol-Eko) and provided with ad libitum food (Labdiet 5008, 3.56 kcal g$^{-1}$) and water purified by reverse osmosis. Mice were maintained under 12/12 h light/dark photoperiods (06:00–18:00) at an ambient temperature of $23 \pm 0.2$ °C. Temperature transitions from 23 to 4 °C were performed over 3 h, starting at 06:00. Data were exported with Macro Interpreter, macro 13 (Sable Systems) before analysis in CalR v.1.3 (ref. 40). Rates of energy expenditure were calculated using the Weir equation[41]. Two datasets were analysed. Dataset 1: 24 mice were maintained at 23 °C for 2.5 days (62 h). A 4 °C ambient temperature was maintained for an additional 9 h before returning to 23 °C for 16 h. Dataset 2: 11 mice were maintained at 23 °C for 5 days (115 h) followed by 4 °C for a further 5 days (118 h) (Extended Data Fig. 1c–e).

### vCAPTURE labelling of Xi$^{CIEC}$-associated neurons

To label CIEC-associated neurons in the vMT/XI region with activating Gq-DREADD, mice received a stereotaxic injection of viral cocktail with ESARE-ER-Cre-ER + DIO-Gq and were habituated with the setup every day for 2–3 h for 1 week. ESARE-ER-Cre-ER/Gq mice were randomly

divided into Xi[CIEC] and Xinon-CIEC groups. On the day of labelling, the Xi[CIEC] group of mice were placed in cold conditions for 6 h with ad libitum food and water. Mice were given 20 mg kg$^{-1}$ 4-TM (Sigma, no. H6278) by intraperitoneal injection 6 h after being placed under cold conditions. When these mice enter CIEC, the vMT/Xi neurons that are active during CIEC induce FOS and CreERT2. Following 4-TM injection, CreERT2 will selectively recombine the AAV8-hSyn-DIO-Gq–mCherry-encoding Cre-dependent Gq-DREADD in the cell that was expressing FOS. This recombination leads to permanent expression of Gq-DREADD in those cells, which can then be selectively activated at a later point by administration of the DREADD agonist CNO. Following 4-TM injections, these mice were kept under cold conditions for a further 2 h before being returned to their home cage and maintained at 23 °C. As a control, we generated a further cohort of mice that received a stereotaxic injection of the same viral cocktail as Xi[CIEC] and were habituated using the same protocol except that, on the day of labelling, they received 20 mg kg$^{-1}$ 4-TM injection after being maintained at 30 °C for 6 h rather than being kept under cold conditions. The c-Fos data in Fig. 1 show that very few cells were active in the vMT/Xi region while mice were under thermoneutral conditions for 6 h compared with the scenario under cold conditions. We term this cohort of mice Xinon-CIEC. Both sets of mice were injected at the same time of day to avoid labelling variability associated with the circadian rhythm. A similar protocol was followed for the optogenetics mice that received a stereotaxic injection of a viral cocktail of AAV5-ESARE-ER-Cre-ER-PEST and AAV8-Ef1a-DIO-hChR2(H134R)-p2AScarlett.

## Histology

Mice were anaesthetized with isoflurane before transcardial perfusion with ice-cold PBS, followed by cold 4% paraformaldehyde (PFA). Brains were removed and post-fixed overnight with 4% PFA at 4 °C. The following day, fixed brains were submerged in 3% agarose and kept under cold conditions for 2 h for embedding. Brains were sliced on a vibratome (Leica VT1000S) into 80 μm coronal sections and collected as two equal sets in a six-well plate filled with PBS. For immunohistochemistry staining permeabilization, brain sections were incubated with a blocking buffer containing PBS, 0.3% TritonX-100 (PBST) and 5% donkey serum for 1 h at room temperature, with gentle shaking. After permeabilization, brain sections were incubated with primary antibodies at 4 °C in a blocking buffer for 16 h. Next, brain slices were washed three times with PBST for 20 min each to wash away unbound primary antibody and were then incubated with secondary antibodies for 1 h at room temperature diluted in blocking buffer. Finally, brain slices were washed again with PBST three times, 20 min each, to wash away unbound secondary antibody before mounting onto super-frost Plus glass slides (VWR). Brain slices were imaged using an Olympus FV3000 confocal microscope with ×10 objective, 0.6 numerical aperture and water immersion (XLUMPlanFI, Olympus). Each brain slice was imaged at 10 μm Z-stack. c-Fos quantification was performed for a single plain (numbers mm$^{-2}$). For double-labelling c-Fos vCAPTURE quantification, cells were counted in a region of interest defined manually following anatomical landmarks. Antibodies used for histology are: c-Fos antibody (Cell signaling, no. 2250, diluted 1:400), anti-rabbit-488 (Jackson Immuno Research, no. 711-546-152, diluted 1:400).

## Whole-brain imaging and analysis

Wild-type C57BL/J6 mice were habituated to the behavioural setup for 5 days before the actual experiment. On the day of the experiment, individually housed mice were maintained under either cold (4 °C) or thermoneutral (30 °C) conditions for 6 h with free access to food and water before perfusion. Brains were harvested and fixed overnight in 4% PFA. Whole-brain clearing, imaging and automated analysis were performed by LifeCanvas Technologies through a contracted service. In brief, fixed whole brains were prepared with SHIELD (stabilization under harsh conditions via intramolecular epoxide linkages to prevent degradation) to preserve protein antigenicity before being actively cleared and immunolabelled with a c-Fos antibody using SmartBatch+. Labelled brains were index matched by EasyIndex and imaged by volumetric lightsheet microscopy (SmartSPIM), followed by image post-processing, cell quantification, atlas registration and regional graphics. The averaged numbers of c-Fos$^+$ cells in each region were grouped into seven 'buckets' based on ABA annotations for data visualization. Dot size represents mean differences between 30 and 4 °C conditions in each region.

## Fibre photometry

Mice were attached to a patch fibre (400 μm core, NA 0.5, RWD) connected to 470 and 410 nm light sources. Fibre photometry and the acquisition setup have previously been described[25]. The 470 nm excitation light was used to measure the Ca$^{2+}$ signal and the 410 nm excitation light as a reference signal. Changes in Ca$^{2+}$ fluorescence (470 nm) signal were compared with background Ca$^{2+}$ fluorescence (410 nm), providing an internal control for movement and signal bleaching. A scientific complementary metal-oxide semiconductor camera (Hamamatsu, Orca Flash 4.0 v.2) was used to capture the patchcord end-face images, and these were processed using a previously described MATLAB code[25]. Data-acquisition hardware (National Instruments, NI PCIe-6343-X) was used to digitize images at 5 kHz. Signals were expressed in d$F/F$, where $F$ represents baseline fluorescence. Video recording of behaviour was used to manually annotate and timestamp feeding and other behaviours. AUC and peak values were calculated for −20/−10 to 10 s from the defined event (for feeding and HMM states, respectively).

For fasting–refeeding experiments, mice were fasted for 16 h. Mice were placed in a tall chamber and their cannula was attached to the fibre photometry setup. Mice were allowed to habituate for 30 min and a feeding device (FED3) was placed at one corner of the chamber. FED3 was programmed to begin dispersing pellets 15 min after placing the device in the chamber to avoid the introduction of noise in the experiment. Data acquisition, analysis and processing were carried out as described above.

## Measurement of serum leptin, glycerol and free fatty acid

Twenty mice were divided into five groups: 1 (2 h, cold), 2 (4 h, cold), 3 (2 h, thermoneutral), 4 (4 h, thermoneutral) and 5 (home cage, 0 h). Quantification of blood plasma measurement was performed according to the manufacturers' recommendations: Mouse Leptin ELISA kit (Crystal Chem, no. 90030), Free fatty acid quantification kit (abcam, no. ab65341) and Free glycerol assay kit (abcam, no. ab65337).

## HMM

Wild-type C57BL/6 mice were habituated to the setup (using FED2) for 4 days before recording their behaviours at 4 °C. For each mouse two videos were recorded, one from the top view and one from the side, using two separate Logitech C270 web cameras. We manually annotated the start and end of each behaviour. Analysers were blind to the experimental groups of mice. Behaviours included sitting, shivering, head grooming, turning, lower body grooming, moving out, eating, moving back, pushing bedding, standing up, bedding retrieval, drinking, digging, grooming tail and walking. Behaviours were annotated every 1 s. HMM requires an input sequence, an estimated transition matrix and an estimated emission matrix. The input sequence was generated with data from the 180 min videos, previously analysed manually, of mice exposed to cold conditions for 4–6 h with free access to food and water. The input sequence of each mouse was put into a vector. We defined a three-state HMM: (1) energy-conserving state, (2) exploration with food consumption and (3) exploration without food consumption. We generated the initial guess for the estimated transition matrix by assuming equal probabilities for all transitions, because we did not have a priori information. We generated the initial guess for the estimated emission matrix by predicting the probability

of behaviour in each state. We used a custom MATLAB code using the Baum–Welch algorithm to acquire transition and emission matrices. To calculate the underlying state of behaviour based on a sequence of behaviours, we used the MATLAB built-in function hmmviterbi.

## RTPP

For basal place-preference measurements of activity-dependent optogenetic mice, either ChR2-XiCIEC or RFP-Xi[CIEC] mice were placed in a two-chamber acrylic box ($60 \times 25 \times 30$ cm³), with each side of the chamber measuring $30 \times 25$ cm², at room temperature (23 °C). Each chamber had different contextual pattern cues on the wall: one side had a black-and-white-striped pattern and the other a dotted pattern. Mouse movements were recorded with an overhead top-view Logitech web camera for the entire 30 min session. Mice were habituated for several hours daily with a fibreoptic cable attached to their head implants for 1 week in a tall chamber before basal place preference. Videos were manually quantified for the amount of time spent in each chamber. For measurement of RTPP at room temperature, mice were placed in the centre of the place-preference chamber and the laser was switched on when all four paws of the mouse were in the stimulation chamber. The laser was manually switched on (10 mW, 10 ms, 20 Hz) and off by the operator sitting in the next room while monitoring a live overhead video of the mouse. The laser was kept on while mice remained in the stimulation chamber and switched off when they left. To test the change in valence while mice were undergoing CIEC, they were maintained at 4 °C for 5–6 h with access to ad libitum food and water before performing RTPP under cold conditions. Mice were given a 1 week gap between RTPP at 23 and 4 °C.

## Open-field test

To measure anxiety-like behaviour in the optogenetics cohort, mice were placed in the outer zone of a white-walled, acrylic, open-field arena ($60 \times 60$ cm²) and movement was recorded for 20 min using a top-view Logitech web camera. The laser was turned on and off intermittently for a 3 min period by an operator sitting in the next room while monitoring a live video feed. Total time and distance in the central 40% of the area were analysed using custom automated tracking software. The open-field test was performed only once for each mouse. Mice were returned to their home cage after the session.

## Statistics

Two-way ANOVA was used to assess how behaviour was affected by other factors (for example, RTPP tested by optogenetic manipulations and temperature). Unpaired $t$-testing was used for comparisons between two groups. Two-tailed tests were used throughout, with $\alpha = 0.05$. Multiple comparison adjustments, biological replicates and significance definitions are included in each figure legend.

## Reporting summary

Further information on research design is available in the Nature Portfolio Reporting Summary linked to this article.

## Data availability

All numerical data are included in Supplementary Information. All other data are too large to deposit in a public repository but are available from corresponding author.

## Code availability

Code used in this study can be accessed at Zenodo, https://doi.org/10.5281/zenodo.7869467.

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

**Acknowledgements** We thank J. Read, A. Brashears and A. Kanow for technical assistance; H. Wang, H. Zhu and E. Zorrilla for the initial behavioural setup; Y. Wang for illustrations; Scripps Research Department of Animal Resources for animal resources; and members of the Ye laboratory for discussions. We thank L. Stowers, C. Kim, V. Augustine and W. Hong for feedback on the manuscript. We thank A. V. Kravitz for FED3 applications and H. Bito for the ESARE plasmid. N.K.L. was supported by the AHA Postdoctoral Fellowship (20POST35200001), Dorris Scholar Award and the Eric and Wendy Schmidt AI in Science Postdoctoral Fellowship. P.L. was supported by the Schmidt Science Fellowship and Eric and Wendy Schmidt AI in Science Postdoctoral Fellowship. T.Q., Z.P. and D.Y. were supported by the Dorris Scholar Award. L.Y. is supported by the National Institutes of Health Director's New Innovator Award (no. DP2DK128800), NIDDK (nos. DK114165, DK134609 and DK124731), BRAIN Initiative/NIMH (no. MH132570), the Dana Foundation, the Whitehall Foundation, the Baxter Foundation and the Abide-Vividion Endowment. A.S.B. is supported by award nos. DK133948, DK107717 and OD028635.

**Author contributions** Conceptualization was carried out by N.K.L. and L.Y. Investigation and analysis were performed by N.K.L., P.L., S.A., A.Z., K.W., T.Q., Z.P., D.Y., V.N. and A.S.B. Indirect calorimetry was performed by A.S.B. Modelling and computational analysis was performed by P.L. and G.W.Y. Funding acquisition was the responsibility of L.Y. Project administration was carried out by L.Y. Supervision was undertaken by L.Y., G.W.Y. and A.S.B. Writing of the original draft was by N.K.L. and L.Y. Writing, review and editing were contributed to by all authors.

**Competing interests** The authors declare no competing interests.

**Additional information**
**Correspondence and requests for materials** should be addressed to Li Ye.

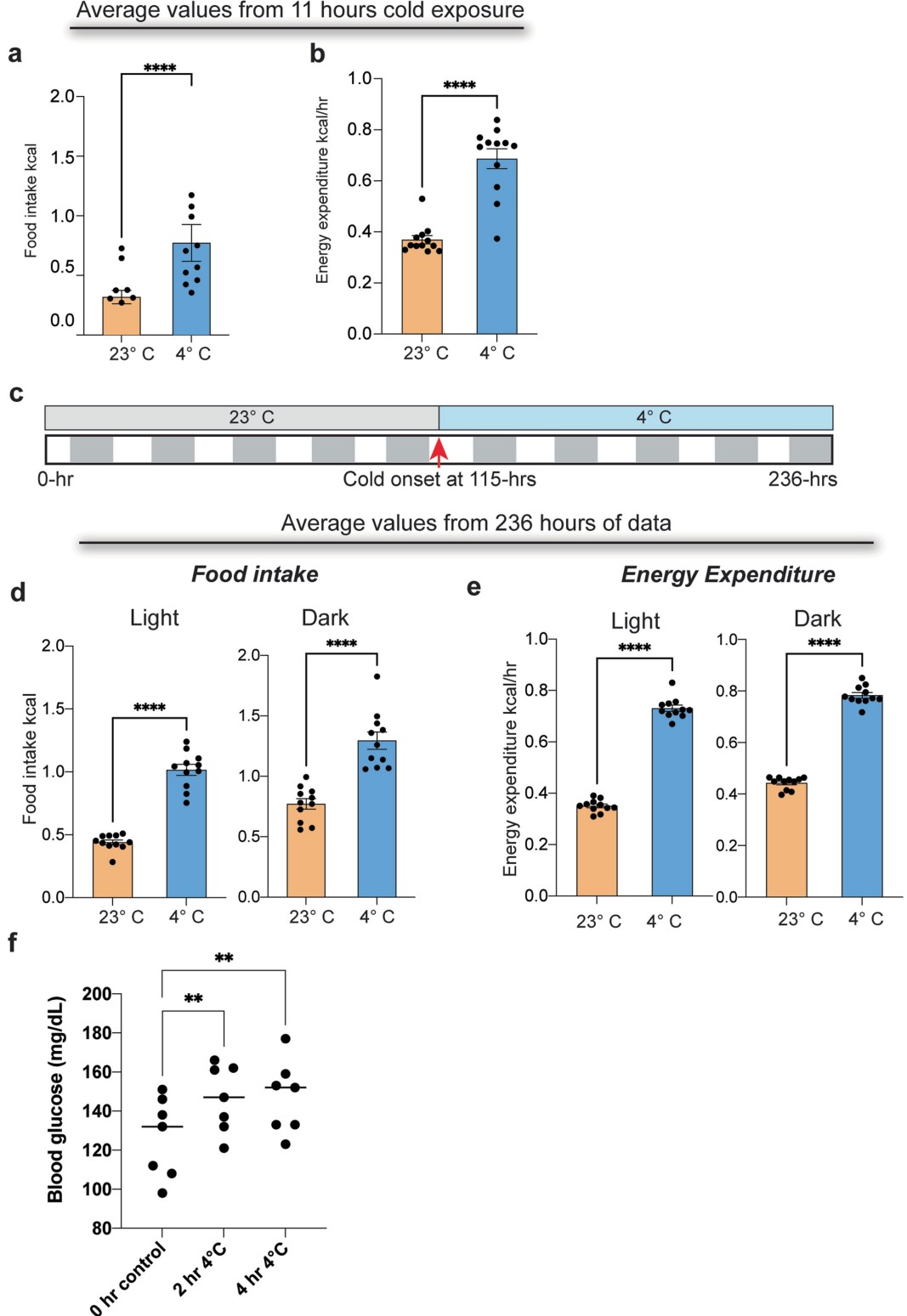

**Extended Data Fig. 1 | Metabolic changes during CIEC. a–b**, Bar graphs showing average food intake (n = 24 mice) (**a**) and energy expenditure (n = 24 mice) (**b**) for data presented in Fig. 1a–c. c. Schematic depicting the long-term cold exposure protocol. 5 days (115 h) after being at RT (23 °C), the temperature was switched to 4 °C. white: light phase, grey: dark phase. Data are mean ± SEM. ****P < 0.0001 using two tailed paired t-test. **d–e**, Quantification of food intake (n = 24 mice) (**d**) and energy expenditure (n = 24 mice) (**e**) during light and dark phases before and after the temperature switch. For calculating the average food intake and energy expenditure, values from either 4 days or 4 nights were used for both RT and 4 °C (excluding the day of temperature switch). Data are mean ± SEM. ****P < 0.0001 using a two tailed paired t-test. **f**, Blood glucose levels of WT mice at 2 h and 4 h after exposure to cold (timepoints are from the same animals) (n = 8 mice). Data are mean ± SEM. **P = 0.001 for comparison between 0 hr and 2 h 4 °C, **P = 0.0028 for 0 h vs 4 h 4C, using a one-way ANOVA with Dunnett's multiple comparison test.

**a**

Estimation of Emission Matrix (Percentage)

| | Sit | Shiver | Groom head | Turn | Lower body groom | Move out | Eat | Move back | Push bedding | Stand up | Bedding retrieval | Drink | Digging | Groom tail | Walk |
|---|---|---|---|---|---|---|---|---|---|---|---|---|---|---|---|
| State 1 (Energy conserving) | 87 | 6 | 2.1 | 0.9 | 1.9 | 0 | 0 | 0.57 | 0.54 | 0 | 0 | 0 | 0 | 0.32 | 0 |
| State 2 (Exploration with food seeking) | 0 | 0 | 0 | 0 | 0 | 24.5 | 74.9 | 0 | 0 | 0.58 | 0 | 0 | 0 | 0 | 0 |
| State 3 (Exploration without food seeking) | 0 | 0 | 0 | 0 | 0 | 3.58 | 0 | 0 | 0 | 19.6 | 23.2 | 13.8 | 16.7 | 0 | 23.2 |

**b**

Estimation of Transtion Matrix (Percentage)

| | State 1 (Energy conserving) | State 2 (Exploration with food seeking) | State 3 (Exploration without food seeking) |
|---|---|---|---|
| State 1 (Energy conserving) | 99.56 | 0.21 | 0.23 |
| State 2 (Exploration with food seeking) | 4.71 | 93.55 | 1.75 |
| State 3 (Exploration without food seeking) | 6.50 | 2.22 | 91.28 |

**c**

**Extended Data Fig. 2 | Representation of Hidden Markov model (HMM) of CIEC paradigm in wild-type mice. a**, Using 15 different labels of actions taken by mice in the CIEC paradigm, a HMM estimated the emission matrix with three states (energy conserving (state 1), exploration with food-seeking and consumption (state 2), and exploration without food-seeking (state 3)). (n = 3 mice). **b**, Corresponding transition matrix probabilities between each state. **c**, Transition matrix of different actions taken by the mouse during the CIEC state.

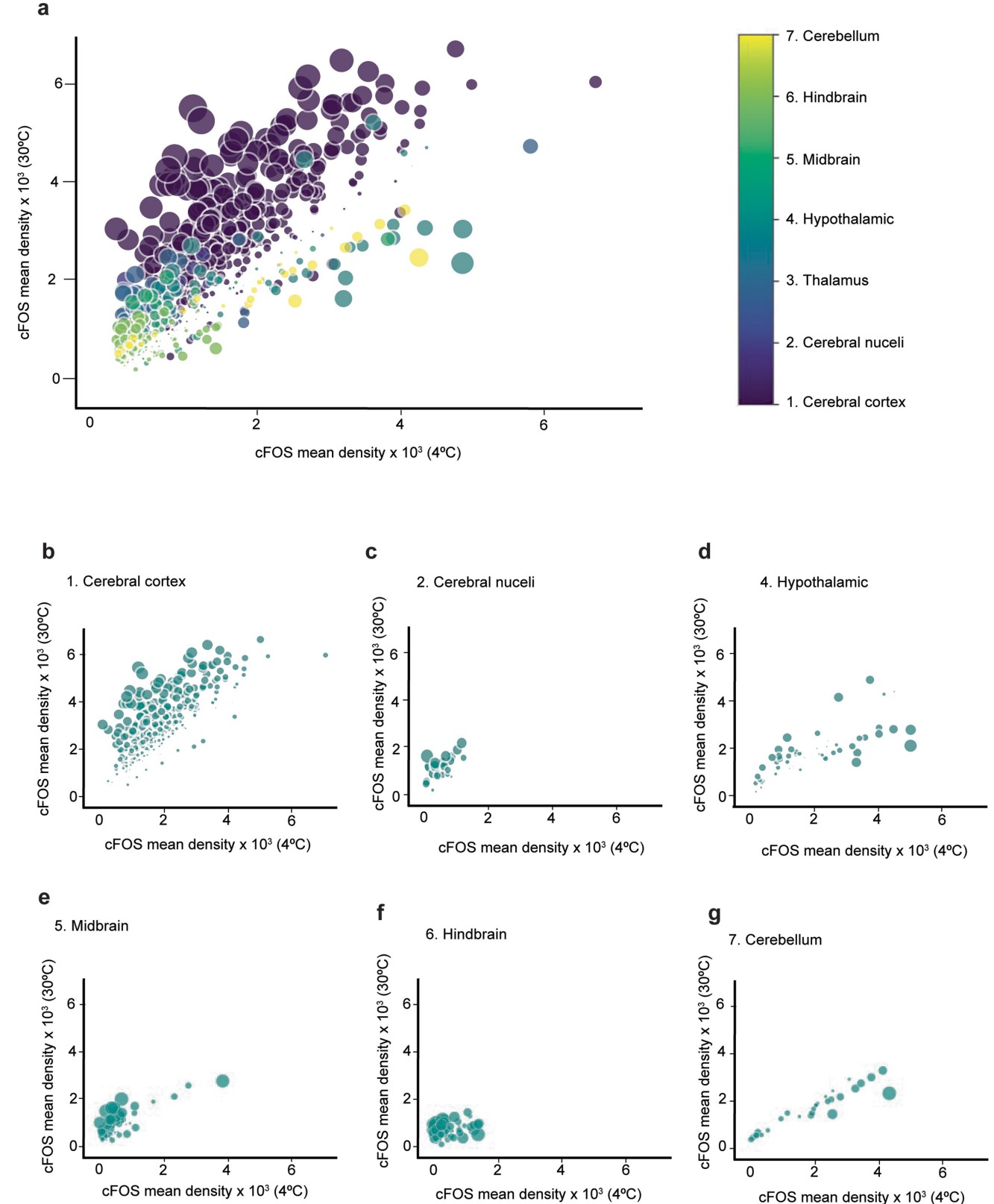

**Extended Data Fig. 3 | Whole-brain screening of mice in non-CIEC 30 °C or CIEC 4 °C state. a–g,** Brains were harvested from mice undergoing CIEC (6 h at cold 4 °C) or non-CIEC (6 h at thermoneutral 30 °C). n = 4 animals per group. Whole-brain imaging and cFos mapping data for the entire brain in **a** and individual regions of the cerebral cortex in **b**, cerebral nuclei **c**, hypothalamus **d**, midbrain **e**, hindbrain **f**, and cerebellum **g**. Each dot represents an annotated brain region based on the Allen Brain Atlas. The size of the dots represents the mean differences between warm and cold conditions in each region. Also see supplementary table 1 for all regions. See also Supplementary Table 1.

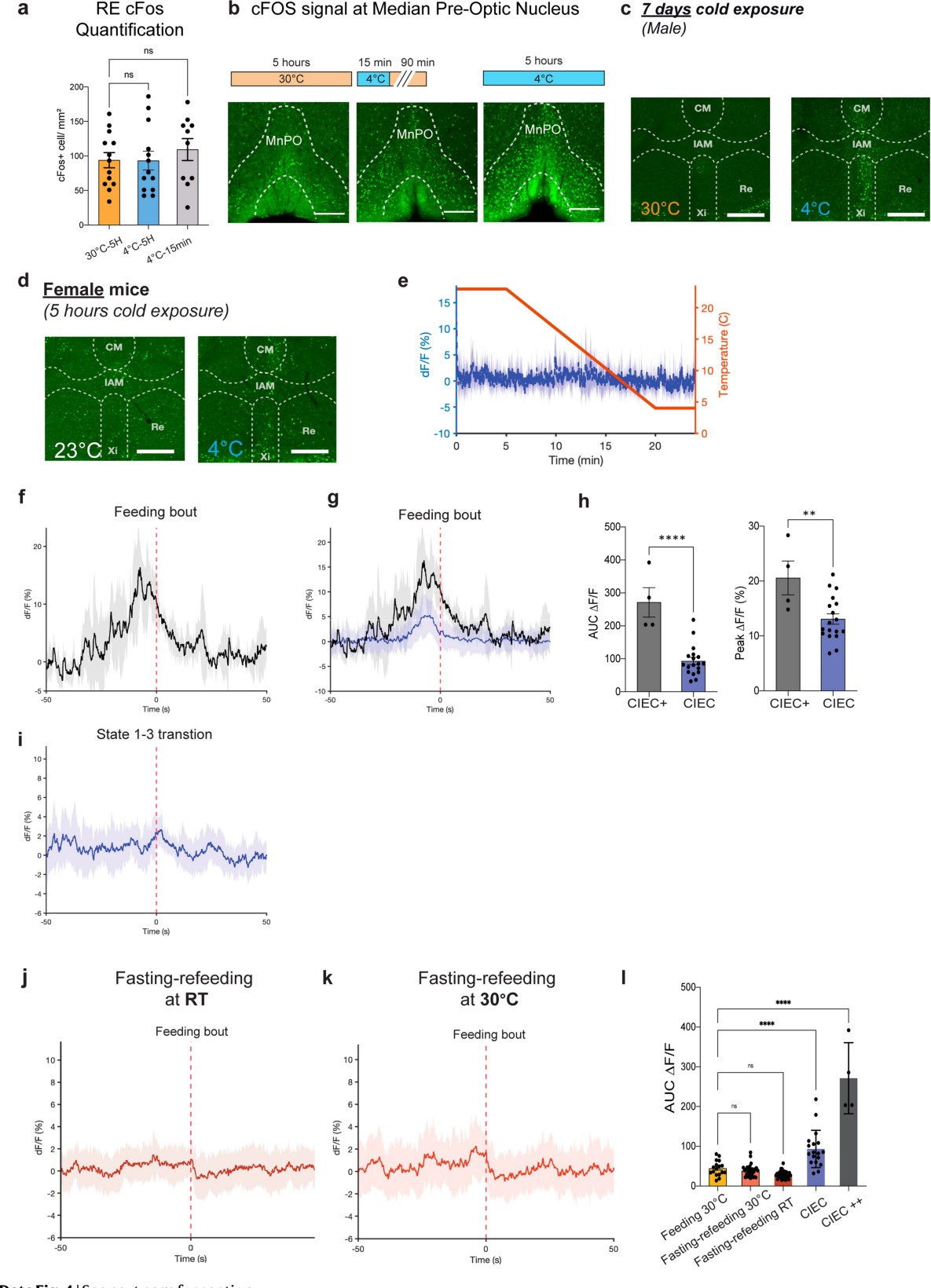

**Extended Data Fig. 4** | See next page for caption.

**Extended Data Fig. 4 | Activity for Xi neurons during cold sensation vs. CIEC and HMM state transitions. a**, cfos quantification of Re region from main Fig. 2b. (n = 4 mice for each condition). Data are mean ± SEM. ns, non-significant using Ordinary one-way ANOVA with Dunnet's multiple comparison test. **b**, representative images of MnPO (at Bregma 0.14) for same experiment as shown in main Fig. 2a–c. **c**, Representative Xi images from mice after 7 days of cold exposure (n = 4 for RT and cold). **d**, Representative Xi c-Fos images from female mice after 5 h of cold exposure. (n = 2 for RT, n = 3 for cold). Scale bar: 200 μm. **e**, Average calcium signal from Xi neurons while the ambient temperature ramped down from 23 °C to 4 °C. (averaged from n = 3 mice). **f**–**g**, Xi calcium signal for mice undergoing CIEC or a CIEC+ state (4 °C without food for 3 hrs to generate exacerbated cold-induced energy compensation), dotted red line indicates a single feeding bout. (n = 4 mice). Solid line is average and shaded area is SEM. **h**, Quantification of AUC delta F/F and peak delta F/F.

Data are mean ± SEM. ****P < 0.0001, **P = 0.01 using two tailed unpaired t-test. (n = 4 mice) **i**, Averaged calcium signal from Xi neurons from mouse undergoing CIEC, averaged from 11 events from 3 different mice. Solid line is average and shaded area is SEM. The red dotted line is the state transition time point calculated using HMM. **j**–**k**, Fiber photometry signal of AAV-GCaMP6m expressing vMT/Xi neurons after overnight fasting during refeeding at room temperature (**j**) or at 30 °C (**k**), shown as the average of 35 events from 5 different mice for (**j**) and 31 events from 5 different mice for (**k**). Solid line is average and shaded area is SEM. **l**, Bar graph showing the area under the curve (AUC) dF/F (−20 s to 10 s) for **j,k** and main Fig. 2f, g. Data are mean ± SEM. ****P < 0.0001, ns = 0.8572 for comparison between 30 °C feeding and fasting-refeeding at 30 °C, ns = 0.1678 for comparison between 30 °C feeding and fasting-refeeding at 23 °C using a one-way ANOVA with Dunnett's multiple comparison test.

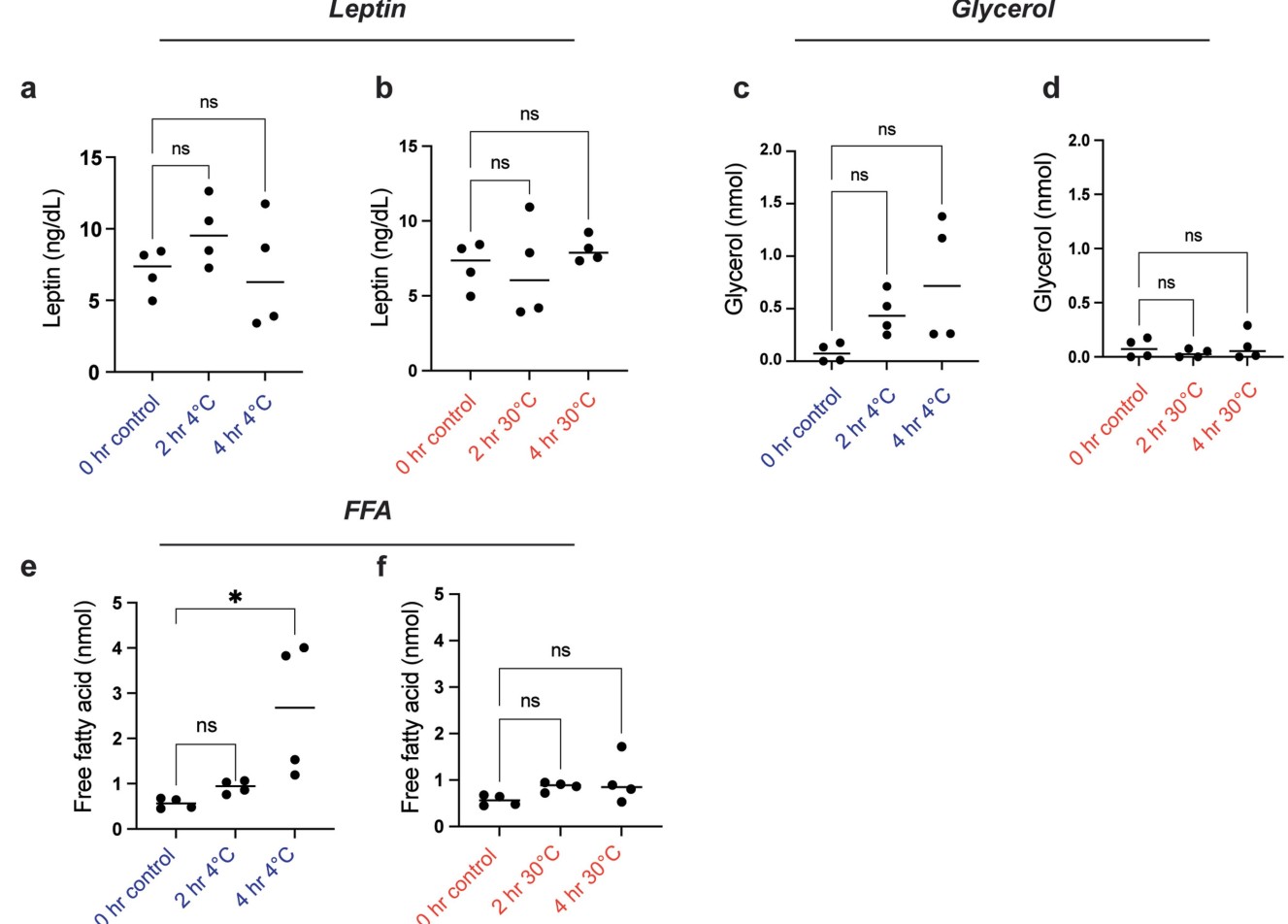

**Extended Data Fig. 5 | Hormonal and metabolic profiling of mice undergoing CIEC vs mice exposed to thermoneutral conditions. a–b,** Quantification of circulating leptin levels in mice after being in cold **(a)** or at 30 °C (thermoneutral) **(b)** for given amount of time (n = 4 mice). **c–f,** Quantification of plasma glycerol (**c** and **d**) and free fatty acids (**e** and **f**) for mice in cold or or at 30 °C (thermoneutral) (n = 4 mice). Data are mean ± SEM. *P = 0.0143 for FFA in cold using a one-way ANOVA with Dunnett's multiple comparison test.

**a**

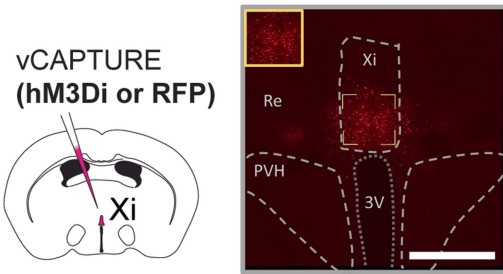

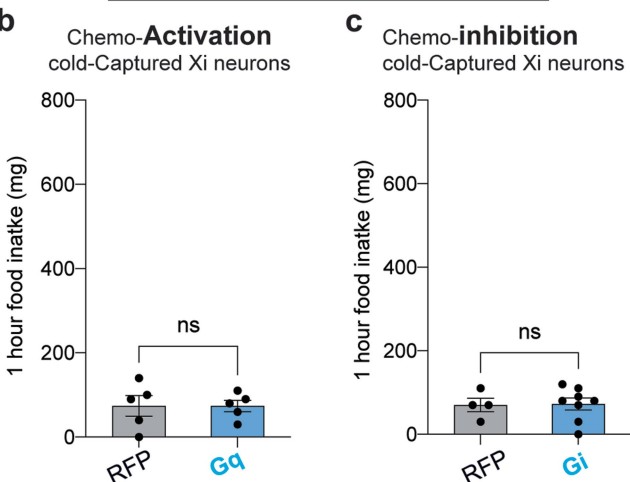

**b**

Chemo-**Activation**
cold-Captured Xi neurons

**c**

Chemo-**inhibition**
cold-Captured Xi neurons

**Extended Data Fig. 6 | Reactivation of cold- vCAPTURE DREADD Xi neurons at room temperature. a**, Schematic and representative histology section from vCAPTURE DREADD Gi mouse. Scale bar: 500 µm. **b**, Food intake for cold-vCAPTURED Xi Gq activated at 23 °C (n = 5 for RFP and Gq). **c**, Food intake data for cold-vCAPTURED Xi Gi at 23 °C (n = 4 for RFP and n = 8 for Gi). Data are mean ± SEM. ns > 0.99 for Gq, ns = 0.9163 for Gi food intake at 23 °C using two tailed unpaired t-test.

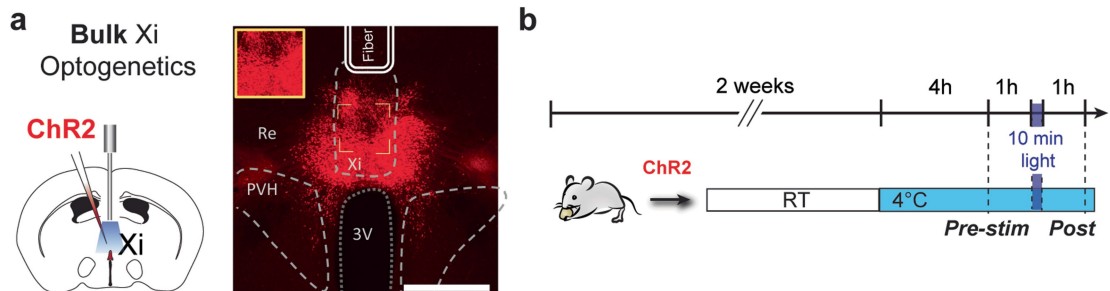

## Bulk optogenetic stimulation of Xi at **4°C**

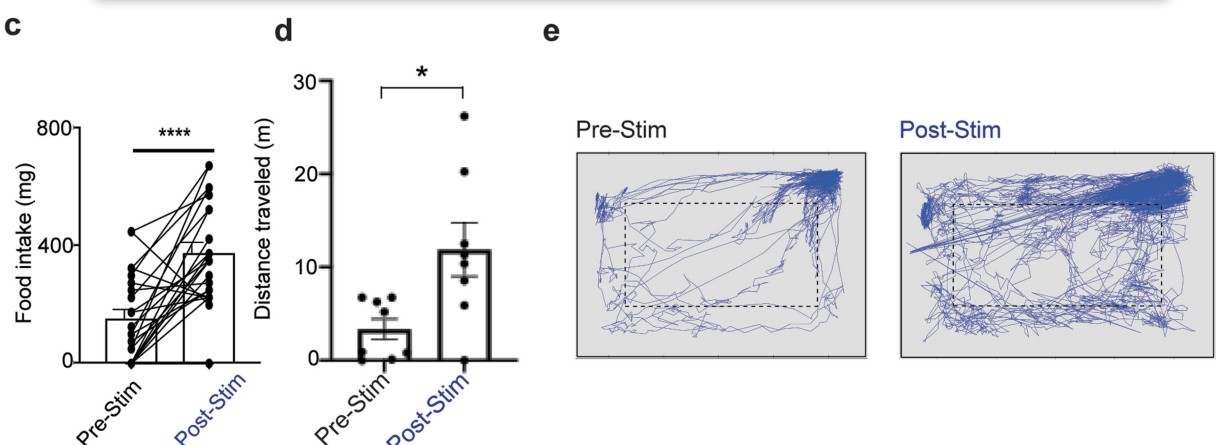

## f Bulk optogenetic stimulation of Xi at **RT**

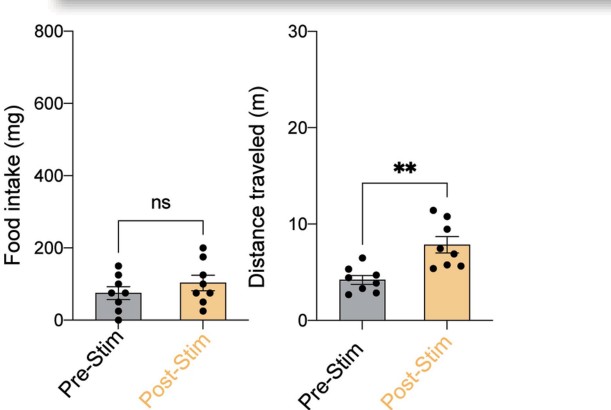

**Extended Data Fig. 7 | Constitutive optogenetic activation of Xi neurons during CIEC. a**, Schematic and representative histology section from a mouse injected with constitutive (hSyn-ChR2) at the Xi. Scale bar: 500 μm. **b**, Experimental design for testing the effect of bulk activation of Xi and surrounding neurons on CIEC-associated feeding. **c–e**, Food intake pre- and post-stimulation (n = 8 mice, N = 16 times) (**c**) and **d–e** physical activity (measured as total distance traveled) pre- and post-stimulation (**d**) (n = 8 mice) and individual traces of physical movement of a single animal (**e**). Data are mean ± SEM. ****P < 0.0001 and *P = 0.0184 using two-tailed paired t-test. **f**, Food intake and physical activity of animals expressing constitutive ChR2 in the Xi at RT (n = 8 animals) Data are mean ± SEM. ****P < 0.0001, **P = 0.0096 using two-tailed paired t-test.

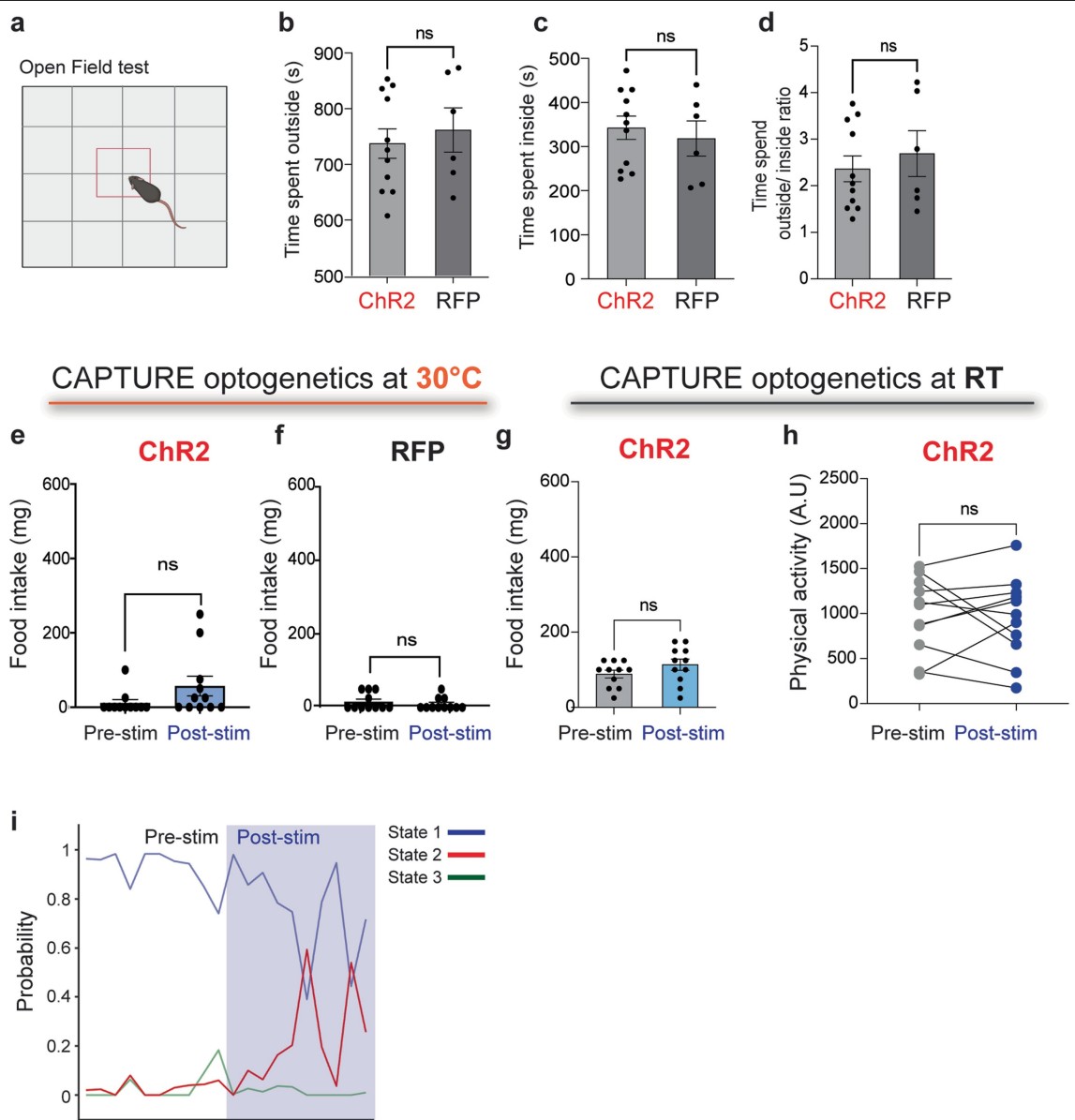

**Extended Data Fig. 8 | vCAPTURE Xi-CIEC optogenetics, open field test, and HMM state transition. a**, Schematic of the open field test. **b**–**d**, Quantification of time spent outside (**b**) or time spent inside (**c**) (or the ratio thereof (**d**) following activation of vCAPTUREd Xi^CIEC neurons in ChR2 or RFP control mice (n = 11 for ChR2 and n = 6 for RFP). Data are mean ± SEM. ns = 0.6087 for (b) and (c), ns = 0.5380 for (d) using two tailed unpaired t-test. **e**–**f**, Bar graph showing the difference in food intake at thermoneutral temperature pre- and post-laser stimulation for ChR2 mice in **e** (n = 11) and RFP mice in **f** (n = 11). Data are mean ± SEM. ns = 0.1501 for **e** and ns = 0.4650 for **f** using two tailed paired t-test. **g**–**h**, Food intake (**g**) and physical activity (**h**) for vCAPTURED Xi^CIEC mice stimulated at room temperature (n = 11 mice). Data are mean ± SEM. ns = 0.1530 for **g** and ns = 0.7614 for **h** using two tailed paired t-test. **i**, Continuous state probability calculated for ChR2 mouse, pre-stimulation (white background) or post-stimulation (blue background) while undergoing CIEC (n = 7 mice).

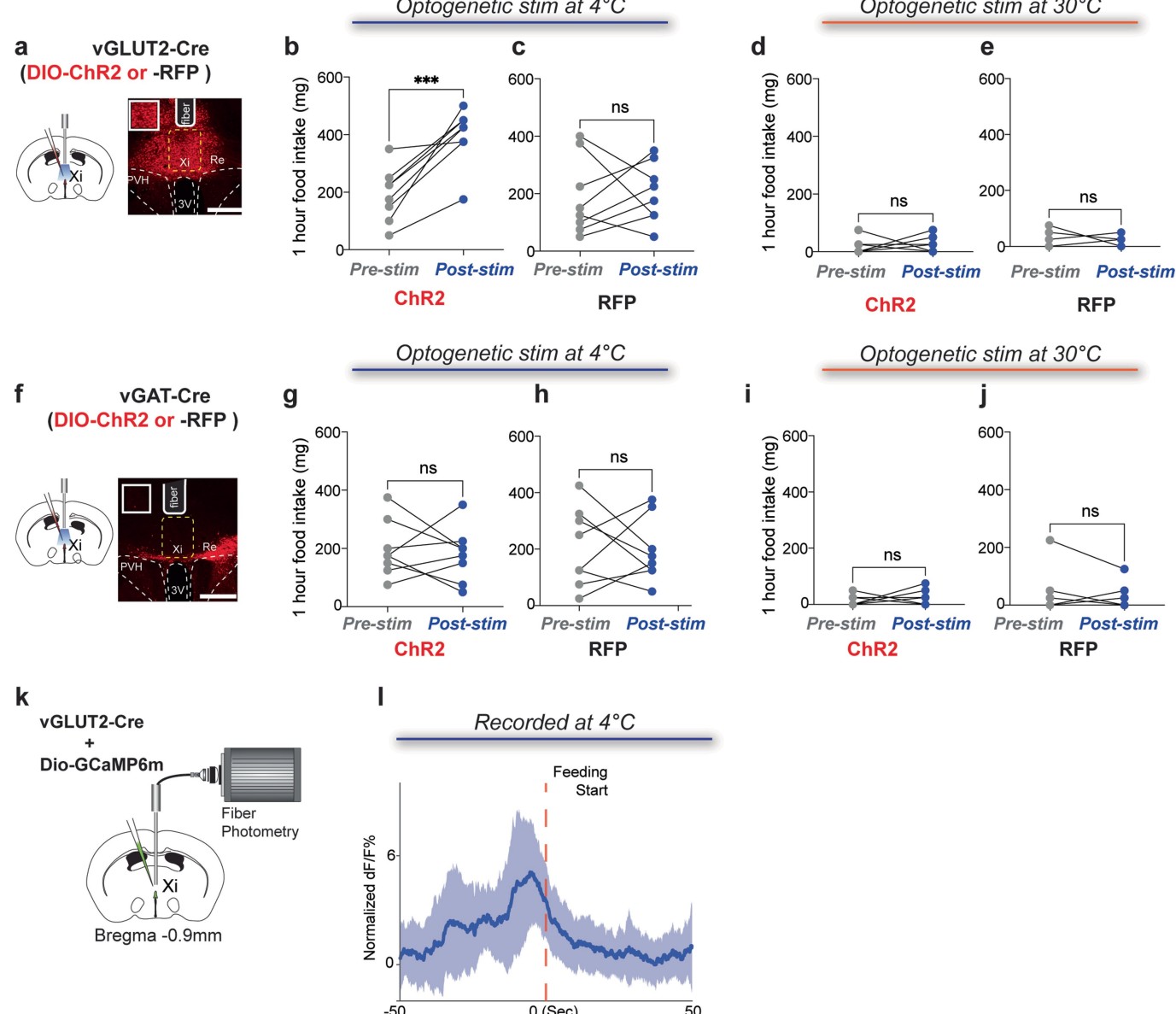

**Extended Data Fig. 9 | Xi$^{vGLUT2}$ neurons mediate CIEC-associated behavior.**
**a**, Schematic and representative histology showing injection of a viral AAV-DIO-ChR2 and optogenetic implant at the Xi (solid white line indicates fiber track) in vGLUT2 animals. Scale bar: 500 μm. **b**–**c**, Food intake in cold pre- and post-stimulation for vGLUT2-ChR2 mice (**b**) (n = 8 mice) and for vGLUT2-RFP control mice (**c**) (n = 8 mice). **d**–**e**, Food intake for vGLUT2-ChR2 mice (**d**) and vGLUT2-RFP mice (**e**) pre- and post-stimulation under thermoneutral conditions. **f**, Schematic and histology of vGAT2 animals. Scale bar: 500 μm. **g**–**j**, Food intake in cold pre- and post-stimulation (**g**–**h**), and under thermoneutral

conditions (**i**-**j**). Data are mean ± SEM for **b**-**j**. ***P = 0.009 for **b**, ns = 0.7849 for **c**, ns = 0.4869 for **d**, ns = 0.8018 for **e**, ns = 0.6682 for **g**, ns = 0.8383 for **h**, ns = 0.5490 for **i**, ns = 0.47 for **j** by two tailed paired t-test. **k**, Schematic of fiber photometry setup for recording from vGLUT2-Xi injected with Dio-GCamp6m at the Xi in vGLUT2 animals. **l**, Fiber photometry signal of AAV-Dio-GCaMP6m expressing in vGLUT2-Xi neurons in the cold, shown as the average of 15 events from 4 different mice. Solid line is average and shaded area is SEM.

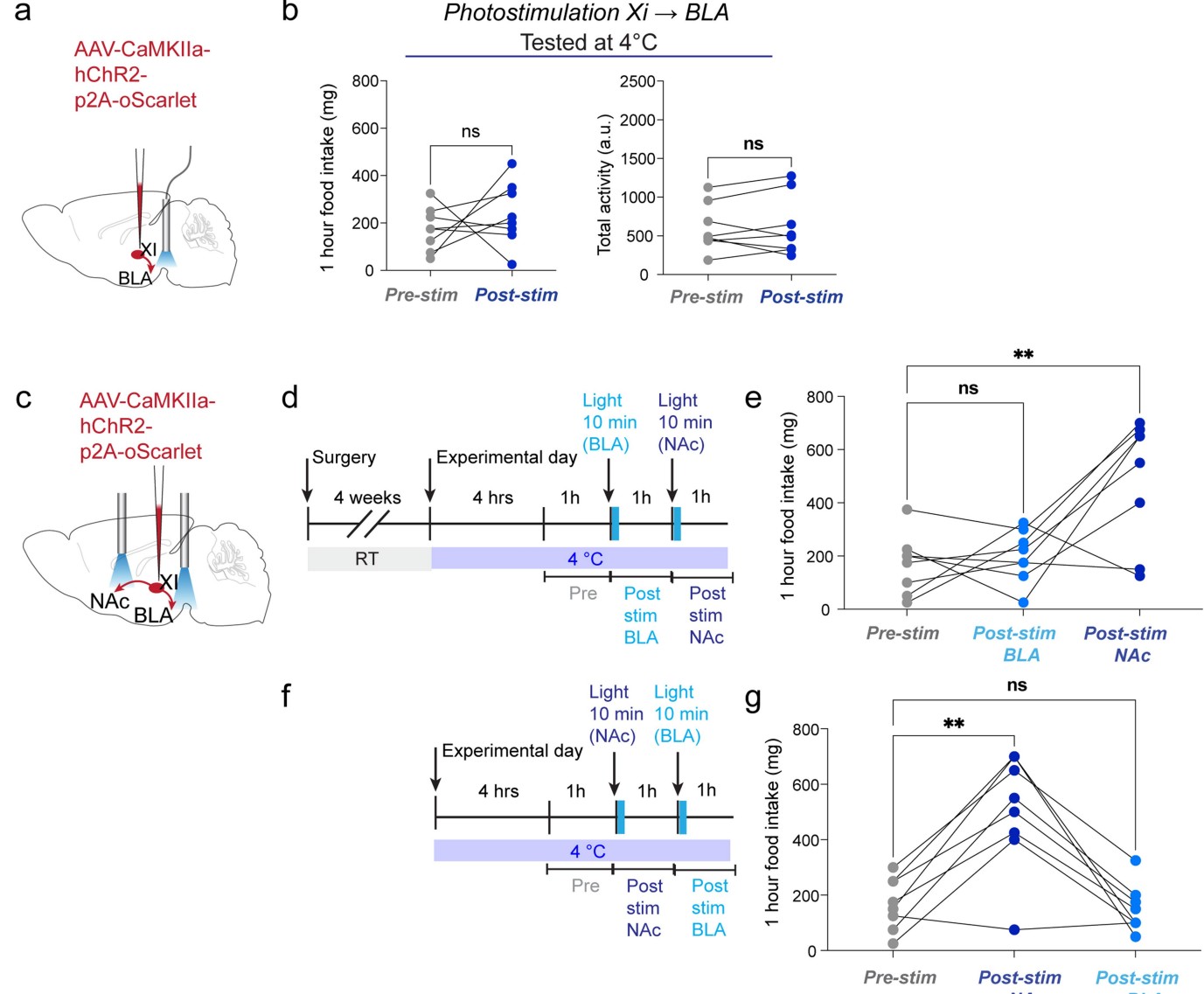

**Extended Data Fig. 10 | Xi-NAc but not Xi-BLA regulate CIEC-associated feeding. a**, Schematic of viral AAV-ChR2 injection at the Xi and optogenetic implant above the BLA in WT animals. **b**, Food intake and physical activity in cold pre- and post-laser stimulation for Xi-BLA mice (n = 8 mice). Data are mean ± SEM, ns = non-significant using two tailed paired t-test. **c**, Schematic of AAV-ChR2 injection at the Xi and optogenetic implant above both the BLA and NAc in the same animal. **d**–**e**, Timeline (**d**) and food intake data (**e**) for the

experiment in which the BLA was stimulated first and then the NAc. Food intake is plotted for pre-stimulation, post-stim for the BLA, and post-stim for the NAc. **f**–**g**, Timeline (**f**) and food intake data (**g**) for experiment in which the NAc was stimulated before the BLA. Data are mean ± SEM. **P = 0.0063, ns = 0.8083 for **e**, **P = 0.0028, ns = 0.7924 for **g** using a one-way ANOVA with Dunnett's multiple comparison test.

# Reporting Summary

## Statistics

For all statistical analyses, confirm that the following items are present in the figure legend, table legend, main text, or Methods section.

| n/a | Confirmed | |
|---|---|---|
| ☐ | ☒ | The exact sample size (*n*) for each experimental group/condition, given as a discrete number and unit of measurement |
| ☐ | ☒ | A statement on whether measurements were taken from distinct samples or whether the same sample was measured repeatedly |
| ☐ | ☒ | The statistical test(s) used AND whether they are one- or two-sided *Only common tests should be described solely by name; describe more complex techniques in the Methods section.* |
| ☐ | ☒ | A description of all covariates tested |
| ☐ | ☒ | A description of any assumptions or corrections, such as tests of normality and adjustment for multiple comparisons |
| ☐ | ☒ | A full description of the statistical parameters including central tendency (e.g. means) or other basic estimates (e.g. regression coefficient) AND variation (e.g. standard deviation) or associated estimates of uncertainty (e.g. confidence intervals) |
| ☐ | ☒ | For null hypothesis testing, the test statistic (e.g. *F*, *t*, *r*) with confidence intervals, effect sizes, degrees of freedom and *P* value noted *Give P values as exact values whenever suitable.* |
| ☒ | ☐ | For Bayesian analysis, information on the choice of priors and Markov chain Monte Carlo settings |
| ☒ | ☐ | For hierarchical and complex designs, identification of the appropriate level for tests and full reporting of outcomes |
| ☒ | ☐ | Estimates of effect sizes (e.g. Cohen's *d*, Pearson's *r*), indicating how they were calculated |

*Our web collection on statistics for biologists contains articles on many of the points above.*

## Software and code

Policy information about availability of computer code

| Data collection | Imaging data for histology was collected using olympur FV31S-SW (version 2.5.1.228. powered by H-PF Version 2.123.2.139).Fiberphotometry data was collected using custom MATLAB (version 2018b) code. Metaboic data was collected with CalR |
|---|---|
| Data analysis | Statistical calculations were performed using GraphPad Prism 9 (GraphPad Software, Inc., La Jolla, CA) and Microsoft Excel 2020. MATLAB (version 2018b) code were used for the analysis of fiberphotometry and HMM data. Behavior data was scored manually using QuickTime 10.4. Imaging data was analyzed using ImageJ(FIJI, 2.3.0).IHC images were prepared for publication using image J (FIJI, 2.3.0). |

For manuscripts utilizing custom algorithms or software that are central to the research but not yet described in published literature, software must be made available to editors and reviewers. We strongly encourage code deposition in a community repository (e.g. GitHub). See the Nature Portfolio guidelines for submitting code & software for further information.

## Data

Policy information about availability of data

All manuscripts must include a data availability statement. This statement should provide the following information, where applicable:
- Accession codes, unique identifiers, or web links for publicly available datasets
- A description of any restrictions on data availability
- For clinical datasets or third party data, please ensure that the statement adheres to our policy

All numerical data is included in the supplementary information. All other data is too large to deposit in a public repository and is available from corresponding author. Source data is provided with the manuscript. Code used in this study can be accessed at Zenodo https: / / zenodo.org / record / 7869467#.ZEnDEezMLmE. Code D0I: 10.5281 / zenodo.7869467

## Human research participants

Policy information about <u>studies involving human research participants and Sex and Gender in Research.</u>

| | |
|---|---|
| Reporting on sex and gender | *Not applicable* |
| Population characteristics | *Not applicable* |
| Recruitment | *Not applicable* |
| Ethics oversight | *Not applicable* |

Note that full information on the approval of the study protocol must also be provided in the manuscript.

# Field-specific reporting

Please select the one below that is the best fit for your research. If you are not sure, read the appropriate sections before making your selection.

☒ Life sciences          ☐ Behavioural & social sciences          ☐ Ecological, evolutionary & environmental sciences

For a reference copy of the document with all sections, see <u>nature.com/documents/nr-reporting-summary-flat.pdf</u>

# Life sciences study design

All studies must disclose on these points even when the disclosure is negative.

| | |
|---|---|
| Sample size | No analysis were performed in advance to predetermine the sample size. Sample size were based on previously published papers in the literature:<br><br>1) Hrvatin, S., Sun, S., Wilcox, O.F. et al. Neurons that regulate mouse torpor. Nature 583, 115–121 (2020). https://doi.org/10.1038/s41586-020-2387-5<br><br>2) Salay, L.D., Ishiko, N. & Huberman, A.D. A midline thalamic circuit determines reactions to visual threat. Nature 557, 183–189 (2018). https://doi.org/10.1038/s41586-018-0078-2 |
| Data exclusions | Data from 3 mice were excluded from figure 2J based on post-hoc analysis of injection site. Data from 2 mice were excluded from Figure 4 C and E based on post-hoc analysis of injection and implantation site. |
| Replication | All the experiments were repeated more than once. All attempts at replication were successful. |
| Randomization | Animals of same age, sex and weight were randomly assigned into treatment or control group. |
| Blinding | Cell counting, fiber photometry data and HMM analysis was performed by person blinded to experimental condition. For all other experiments experimenter was not blinded because the researcher need to know the site of implantation and injection to exclude failed surgeries. |

# Reporting for specific materials, systems and methods

We require information from authors about some types of materials, experimental systems and methods used in many studies. Here, indicate whether each material, system or method listed is relevant to your study. If you are not sure if a list item applies to your research, read the appropriate section before selecting a response.

## Materials & experimental systems

| n/a | Involved in the study |
|---|---|
| ☐ | ☒ Antibodies |
| ☒ | ☐ Eukaryotic cell lines |
| ☒ | ☐ Palaeontology and archaeology |
| ☐ | ☒ Animals and other organisms |
| ☒ | ☐ Clinical data |
| ☒ | ☐ Dual use research of concern |

## Methods

| n/a | Involved in the study |
|---|---|
| ☒ | ☐ ChIP-seq |
| ☒ | ☐ Flow cytometry |
| ☒ | ☐ MRI-based neuroimaging |

## Antibodies

| | |
|---|---|
| Antibodies used | cFOS antibody (Cell signaling catalog number 2250) (diluted 1:400) , Anti-Rabbit-488(Jackson Immuno Research 711-546-152, dilution of 1:400) |
| Validation | All antibodies were sourced from well established companies and are thoroughly validated by manufacturer and is widely used in neuroscience literature. cFOS antibody (Cell signaling 2250) https://www.cellsignal.com/products/primary-antibodies/c-fos-9f6-rabbit-mab/2250 |

## Animals and other research organisms

Policy information about studies involving animals; ARRIVE guidelines recommended for reporting animal research, and Sex and Gender in Research

| | |
|---|---|
| Laboratory animals | Mice were housed with standard 12 hour light/dark cycle at 23C. The lights were on from 6 am to 6 pm. All experiments were performed on wild type C57BL6/J mice ordered from the Scripps Research Department of Animal Resources rodent breeding colony. Experiments were performed on mice between the age of 2 to 6 months. |
| Wild animals | No wild animals were used. |
| Reporting on sex | The findings reported here apply to both sexes. The initial screening was performed in both sexes. cFOS data for both sexes are reported separately in Fig. 2B and Fig. S4A |
| Field-collected samples | No field-collected samples were used. |
| Ethics oversight | Animal experiments were performed either at Scripps Research Institute, La Jolla or Beth Israel Deaconess Medical Center (BIDMC). Experiments were approved by the Scripps Research Institute's or BIDMC's Institutional Animal Care and Use Committee (IACUC), respectively. All experiments were in accordance with the guidelines from the NIH. |

Note that full information on the approval of the study protocol must also be provided in the manuscript.

