## [Peer Review File · Nature]

Manuscript Title: Xiphoid nucleus of the midline thalamus controls cold-induced food seeking

Reviewer Comments & Author Rebuttals

Reviewer Reports on the Initial Version:

Referees' comments:

Referee #1 (Remarks to the Author):

The authors set out to find the neurons responsible for cold-induced feeding by using a set of unbiased behavioral classification and activity mapping techniques. After finding that the xiphoid nucleus is activated by chronic (five hours) but not acute (15 minutes) of cold exposure, they analyzed the natural activity of these neurons using fiber photometry. This was followed by functional studies in which the effect of reactivation or reinhibition of Xi neurons previously activated in the cold (using the E-sare system) was tested using chemogenetics and optogenetics.

The data are interesting and the identification of a neural substrate specific to a cold-induced feeding drive is of interest. However, the data still leave open the possibility that these neurons play a more general role in sensing an energy-deficit, and then inducing feeding, rather than being specific for cold induced feeding.

They suggest that the Xi only regulates the transition from energy conservation (State 1 in their parlance) to foraging (State 2). Studies of the effect of these neurons under other conditions is necessary to confirm their hypothesis. As below, further experiments exploring the role of these neurons are necessary to confirm the specificity of these neurons for cold induced feeding. For example, the authors could use the E-sare system to evaluate whether inhibition of the Xi neurons alters food intake after a period of acute or chronic food restriction (a protocol for chronic energy deficit is reported in this publication, <https://www.pnas.org/doi/10.1073/pnas.1002271107>).

The authors also do not take the opportunity establish the identity of the Xi neurons. This could be accomplished in a straightforward manner by employing single cell sequencing after marking the cells expressing cre using E-sare. These studies would enable them to confirm whether a single cell population or multiple ones contribute to the response they observe. Since little is known about this nucleus this information would enhance the paper and enable subsequent studies of the circuit.

The studies of place preference are not adequately explained. Why is activation of these neurons associated with positive valence? Often food intake during states of negative energy balance is associated with the suppression of negative valence (see Betley, 2015 reporting studies of AGRP neurons). Are the authors suggesting that the transition from State 1 to 2 is intrinsically rewarding? If this were the case, why do the animals spend ~ 93% of the time in State 1?

What follows are some additional comments:

In Figure 1, the authors suggest that the increase in food intake lags behind the cold exposure until such time as an energy deficit is sensed. This raises the question as to what signals the energy deficit? Is there a metabolic signal or a hormonal signal. The authors might consider at least some initial experiments to address this including measures of blood glucose, ketone bodies, free fatty acids, glycerol and hormones such as leptin. Identifying the signal is beyond the scope of

this paper but at a minimum the authors should exclude known signals that might contribute. In addition, the authors need to account for possible circadian effects and should thus provide the same correlation as shown in Panel (C) but on the day prior to exposing the animals to the cold exposure. The data should then be analyzed by comparing the correlations on the day before and after cold exposure.

The fiber photometry experiments (such as Figure 2F,J) would be more informative if the authors could show if the activity shown is specific to cold-induced feeding drive as opposed to a feeding drive due to an acute or chronic energy deficit. The studies of the CIED++ state raise this possibility by showing that an energy deficit before cold exposure enhances the responses.

In the functional studies using chemogenetics to inhibit the previously activated Xi neurons, they should repeat the studies but without starving the animals beforehand. It is unnecessary and possible confounding to fast the animals as they should have elevated food intake during the period of cold exposure. They should also repeat the studies in fasted mice and in chronically malnourished mice at 30C, see <https://www.pnas.org/doi/10.1073/pnas.1002271107>

Inhibition of the Xi captured neurons decreases core temperature but it is unclear whether the reduced food intake leads to the reduced body temperature, as the authors suggest, or if the reduced body temperature decreases food intake. At a lower body temperature, energy expenditure is decreased thus diminishing the energy deficit. The authors should try to distinguish between these possibilities

In the studies of RTPP, the authors should provide statistics comparing place preference at 4C to 30C. .

Overall the paper reports potentially interesting findings but further studies are required to confirm their conclusions.

Referee #2 (Remarks to the Author):

This study used a whole brain imaging method to identify key brain areas that are activated in response to cold-induced responses including increased energy expenditure and feeding. Based on a relatively long 4C cold exposure model (11 hr exposure ending on the onset of the dark cycle), it was identified that a few brain sites responded with c-Fos expression including a novel, functionally less known, mid thalamic region called the Xi nucleus. Using an established viral trapping approach, the Xi c-Fos neurons were permanently trapped and studied for its role during cold exposure and it was found that these neurons were active during cold-induced feeding, promoted feeding and transition from state 1 (energy conserving static) to state 2 (exploratory feeding). In addition, activation of these neurons was preferred only during 4C cold exposure.

The results on the Xi neurons in response to cold and promoting feeding behavior is completely novel and exciting, which should be interesting to broad readership. The experimental approaches are rigorous and the results are convincing with appropriate statistical analysis. The analysis of animal behaviors in 4C cold with 3 different states is also helpful in differentiating various physiologic states in cold exposure.

One major big-picture concern is that the results as presented seem to be a bit isolated in nature. What are the upstream and downstream neurons that may be important in mediating the observed effect? Given the large body of literature on various brain neurons that also respond to ambient temperatures and regulate metabolism, any relation of Xi neurons to those known neurons will help to build a big picture. Importantly, one previous study (Deem JD et al., Elife, 2020) reported

a similar role for AgRP neurons in promoting feeding with cold exposure (14C). Is there any relationship between these 2 sets of neurons? Are AgRP neurons among those identified in the whole brain screening?

One potential issue is the alignment of cold exposure with the diurnal pattern (Fig. 1a) used in this study. The 11hr 4C exposure was placed with an ending at the onset of the dark cycle, in which the increases in both energy expenditure and feeding during the onset of the dark cycle coincided with diurnal increases in both, leading to a concern that the observed increase in the late phase of cold exposure may not be due to a direct effect of cold exposure, thus the analysis on the relationship between increased energy expenditure and feeding presented in Fig. 1c may not be as the authors argued. It would be helpful to confirm this finding with the 11hr 4C exposure ending at the middle of the light cycle.

Data presented in Fig. 2a-2c with the 15min 4C exposure is somewhat confusing. This short cold exposure may be too short to induce any changes including c-Fos expression or may be too late (5h later) to see c-Fos expression. Multiple durations of 4C exposure should be used with a shorter delay prior to tissue collection may be required to make the point.

There appears to be inconsistency between data presented in Fig. 1 and Fig. 2. Fig. 2 showed that 5h 4C exposure activated a subset of Xi neurons, which led to increased feeding. However, Fig. 1 data showed that after 5h 4C exposure, the feeding was not increased until at a much later phase of the cold exposure. The authors need to clarify on this.

Fig. 3g data on Gq activation at 30C seem to be confusing. CNO-Gq induced activation of Xi neurons should mimic cold exposure and therefore activation of these neurons should promote feeding regardless of temperature exposed. However, the data presented showing Gq activation at 30C had no difference compared to the control group. Clarifications are needed for this point.

Referee #3 (Remarks to the Author):

The manuscript by Lal et al first performs a phenotypic analysis to determine the time course of stereotyped behaviors of mice during exposure to 4C. Next, they screen for brain Fos activity as a result of cold-induced food intake to identify the Xi as a region differentially activated by cold exposure compared to thermoneutrality. Using calcium monitoring and chemogenetic/optogenetic neural activity manipulations, they then determine that Xi neurons are activated immediately prior to feeding bouts in the cold and not warm, and that activity in these neurons bi-directionally influences cold-induced feeding. Overall, the findings are interesting and the experiments are scientifically sound. The manuscript is also well-written and overall an enjoyable read. The weaknesses of the manuscript are that (1) some findings are preliminary and (2) the manuscript lacks mechanistic insight into how the Xi is involved in cold-induced feeding behavior. The following suggestions would improve the impact of the manuscript:

Major comments:

1) Several comments regarding Figure 1:

- Quantitative and statistical comparisons between 23C and 4C data in Figure 1A-B should be included in the main figure, for both food intake and energy expenditure
- The HMM is a cool way to analyze behavior, but the cold-induced feeding behavior should be compared to behavior at 23C or 30C
- Further, it is difficult to interpret the modeling data as depicted in Figure 1 and the related supplements. More descriptive explanations in text and/or legends would be very useful.
- Since the cold is performed in the light period, why is sleep not included in the model?

2) The unbiased screen indicated many regions, beyond the Xi, that are activated in 4C (compared to 30C). Is the Xi unique, compared to these other differentially activated regions, in its involvement in cold-induced food intake? If neurons in these other regions are monitored and manipulated neurons in the other regions that were activated, do they also respond to or have an effect on cold-induced feeding?

3) The in vivo fiber photometry analyses are preliminary and raise a variety of further questions.

- Although the authors determined that Xi neurons are not responsive to temperature per se, responsivity was only measured during food intake in 4C and 30C. What happens at temperatures in between? Are there gradual increases in activity as temperature drops? Or is there a critical threshold? This kind of analysis, depending on results, could help the authors' speculations about the Xi as a behavioral gate.
- An analysis of how $\Delta F/F$ relates to the size of the feeding bout may provide insight into the nature of the Xi activity. For example, since the calcium transient occurs prior to the bout, is it predictive of the amount of food intake?
- Fiber photometry was performed only after 5 hours of cold exposure. Does Xi neuron activity in response to food intake change in the early cold vs the late (5+ h) cold? In other words, do Xi neurons respond to food only during the period of cold-induced feeding?
- Minor comment: line 122 indicates a correlation, but there are no correlations in the figure
- I appreciate that the authors included an example raw trace in the inserts for the averaged photometry traces.

4) Only about 50% of the vCAPTURE neurons activated by 4C display cold-induced Fos – a relatively low percentage. The authors should report overlap both ways – the % of Fos neurons that are captured, and the % of captured neurons that have Fos. This low overlap raises concerns about the vCAPTURE neuron specificity – the authors should at the very least discuss this or preferably address it experimentally by recording in vivo dynamics in vCAPTURE neurons.

5) For the neural activity manipulations, the experiments were always performed in the cold. What if instead these neurons were activated or inhibited at RT or thermoneutrality? Is food intake affected? These controls are important to determine whether these effects are specific to cold-induced food intake.

6) Related, how specific are the Xi neuron responses to cold-induced food intake? Are they also activated by other kinds of actual or perceived energy deficits (e.g. exercise, glucoprivation?). Addressing these questions would provide more insight into the specificity of the observed effects.

7) In general, while the manuscript identifies Xi as a region mediating cold-induced food intake, there are no mechanistic underpinnings. The increase in energy expenditure precedes the food intake increases, how is this being signaled? Do these neurons receive inputs from BAT? Conversely, inhibiting these neurons prevents the cold-induced increases in temperature – do they connect to BAT via direct or polysynaptic connections?

8) The ns for each experiment widely vary. For experiments with very low n (e.g., 3-5), more subjects should be added.

9) The discussion section lacks depth and could be better used to interpret findings in the context of the literature.

Other comments:

1) Lines 33-35 should include appropriate citations.

2) Figure 1 D/E, it is very difficult to distinguish between all of the colors.

3) On line 150-151, the authors conclude that these neurons are necessary for survival in the cold, but this statement is quite premature as it is based on 1-h measurements of temperature.

4) Line 181, the authors should not report "slightly negative valence" as this is not statistically significant.

5) Extended data Figure 4e,f – x axes labels are CEID rather than CEIC

Referee #4 (Remarks to the Author):

The work by Lal and collaborators assessed the role of the thalamic xiphoid neuronal activation as driver of cold-induced food intake. They showed that 5-6 hours of cold exposure induced neuronal activation in this midline thalamic structure by means of cFos and in vivo Ca²⁺ imaging. Using chemo- and optogenetics they revealed that the activation of neurons in this nucleus play a role in cold-induced feeding with an interesting positive valence. The role of this nucleus in regulating feeding is novel and the approach used is state-of-the-art.

However, there are some issues that need to be addressed, including the time points analyzed and the environmental temperatures used in several experiments. Furthermore, the identification of the phenotype of these neurons in the xiphoid nucleus and their projections fields requires further studies.

Specific comments:

- 1) The authors assessed cold-induced feeding at 5h from the start of the exposure. However, from Fig. 1b a major increase in feeding can be observed from 2-3 hours from the start of the exposure. Thus, a time course on cold-induced cFos should be performed, including earlier and later time points to determine the correlation between the pattern of food intake and Xi activation.
- 2) I would also suggest for the authors to maintain consistency in the temperature used throughout the study. For example, the experiments in Fig. 1 were done switching either from 23°C to 4°C or from 30°C (thermoneutrality) to 4°C.
- 3) For Fig. 1i, (the whole-brain SHIELD, extended data Figure 3), the authors should disclose all of their cFos studies throughout the brain as a resource for others.
- 4) The authors need to determine whether 4°C-induced Xi neuronal activation can induce food intake at RT or 30°C. Moreover, some other areas including ARC and PVN should be investigated upon Xi-activation.
- 5) Line 134, the authors mentioned "CIEC selectively induced cFos in the Xi without activating the surrounding areas (Fig. 2b)", however, an increase in cFos expression can be seen in areas around Xi. These areas should be quantified as well.
- 6) The synaptic activity-responsive element (SARE) is a neuronal activity-dependent enhancer that regulates the induction of the immediate-early gene (IEG) Arc6. Three activity-dependent transcription factors (CREB, MEF2 and SRF) bind to SARE and synergistically drive high expression of downstream genes. An enhanced version of the SARE promoter (E-SARE) drives downstream expression at much higher levels than previous IEG promoters. Thus, would the AAV-cFos-ER-Cre-ER a better viral approach for the experiments in Fig. 3 than the AAV-ESARE-ER-Cre-ER used in this study (since the authors determined cFos expression in Xi neurons)?
- 7) In Fig. 3g, was food measured at 4°C, 30°C or RT? Does the activation of XiCIECneurons induce food intake at 30°C or RT? What about the food intake follow the activation of neurons "captured at 30°C" and tested at RT or 30°C? Similarly, what about food intake of hM4Di at RT?
- 8) Similar to the chemogenetic experiments, in the optogenetic experiments of Fig. 4, how was the food intake and physical activity when Xi cold-captured neurons were activated at RT?
- 9) Regarding Extended data Fig6 (hSyn-ChR2), food intake and physical activity at RT need to be provided.

Author Rebuttals to Initial Comments:

Response to Reviewers

We are delighted that all reviewers are excited about our study, calling the discoveries “*completely novel and exciting*”, “*an enjoyable read*” and approaches and data “*state-of-the-art*”, “*rigors, convincing with appropriate statistical analysis*”. We thank the reviewers and editors for the highly constructive suggestions, for example, on characterizing whether the Xi activity is specific to the cold or fasting state and its relationships with other circuits. We are happy to report that we were able to complete almost all the requested new experiments and included data in the revision.

Referee #1 (Remarks to the Author):

The authors set out to find the neurons responsible for cold-induced feeding by using a set of unbiased behavioral classification and activity mapping techniques. After finding that the xiphoid nucleus is activated by chronic (five hours) but not acute (15 minutes) of cold exposure, they analyzed the natural activity of these neurons using fiber photometry. This was followed by functional studies in which the effect of reactivation or reinhibition of Xi neurons previously activated in the cold (using the E-sare system) was tested using chemogenetics and optogenetics. The data are interesting and the identification of a neural substrate specific to a cold-induced feeding drive is of interest.

We thank the reviewer for summarizing and agreeing on the significance of the study.

However, the data still leave open the possibility that these neurons play a more general role in sensing an energy-deficit, and then inducing feeding, rather than being specific for cold induced feeding. They suggest that the Xi only regulates the transition from energy conservation (State 1 in their parlance) to foraging (State 2). Studies of the effect of these neurons under other conditions is necessary to confirm their hypothesis. As below, further experiments exploring the role of these neurons are necessary to confirm the specificity of these neurons for cold induced feeding. For example, the authors could use the E-sare system to evaluate whether inhibition of the Xi neurons alters food intake after a period of acute or chronic food restriction (a protocol for chronic energy deficit is reported in this publication, <https://www.pnas.org/doi/10.1073/pnas.1002271107>).

We agree that it is important to determine the specificity of Xi neurons during cold-induced feeding and experimentally differentiate their role in classic fasting-induced feeding paradigms. We recorded Xi activity during feeding after overnight fasting at both room temperature and thermoneutrality. We found that as opposed to the increase of activity observed during cold-induced feeding, we did not see the same change in fasting-induced re-feeding (New Extended Data Fig 5), suggesting a specific role in cold-induced feeding rather a general response to energy deficit.

The authors also do not take the opportunity establish the identity of the Xi neurons. This could be accomplished in a straightforward manner by employing single cell sequencing after marking the cells expressing cre using E-sare. These studies would enable them to confirm whether a single cell population or multiple ones contribute to the response they observe. Since little is known about this nucleus this information would enhance the paper and enable subsequent studies of the circuit.

We appreciate the suggestions to establish the identity of Xi neurons. In new Extended Data Fig 10, we showed that the majority of Xi neurons are vGLUT2+ but not vGAT. In new Fig 5 and Extended Data Fig 11, We also thoroughly combined anterograde, retrograde, and functional (recording and manipulation) approaches to establish the projection-defined identity of the cold-activated Xi population. Considering Xi's small size and being poorly characterized in previous literature, it is not feasible to find corresponding scRNAseq data on this region based on the publicly available database or to establish new scRNAseq experiments within the timeframe of this revision.

The studies of place preference are not adequately explained. Why is activation of these neurons associated with positive valence? Often food intake during states of negative energy balance is associated with the suppression of negative valence (see Betley, 2015 reporting studies of AGRP neurons). Are the authors suggesting that the transition from State 1 to 2 is intrinsically rewarding? If this were the case, why do the animals spend ~ 93% of the time in State 1?

We thank the reviewer for this intriguing question. Although optogenetic activation of Xi in the RTPP setting has a state-dependent positive valence, the endogenous Xi activity (and therefore state transition) is subjected to conflicting priorities and is transient in nature (Fig. 2). In other words, although the valence of Xi activity is positive in the cold, it is time-locked to each behavior episode rather than being constantly "ON", whereas the classic AgRP activity is constantly "ON" in the fasted animals. We now added our discussion on the transient nature of state transition/valence which is consistent with the previously described role of Xi in behavioral switching (Ref 24).

What follows are some additional comments: In Figure 1, the authors suggest that the increase in food intake lags behind the cold exposure until such time as an energy deficit is sensed. This raises the question as to what signals the energy deficit? Is there a metabolic signal or a hormonal signal. The authors might consider at least some initial experiments to address this including measures of blood glucose, ketone bodies, free fatty acids, glycerol and hormones such as leptin. Identifying the signal is beyond the scope of this paper but at a minimum the authors should exclude known signals that might contribute.

Identification of molecule that signals energy deficit will be an interesting future direction, and as the reviewer pointed out, is beyond the scope of this paper. Nonetheless, as requested we characterized both 4c and 30c induced changes in blood glucose (new Extended Data Fig 1f), leptin, glycerol, and free fatty acid levels (new Extended Data Fig 6). Constant with the literature on cold physiology, we observed increases in glucose after cold exposure (but no significant changes in circulating leptin), which are distinct from typical fasting responses.

In addition, the authors need to account for possible circadian effects and should thus provide the same correlation as shown in in Panel (C) but on the day prior to exposing the animals to the cold exposure. The data should then be analyzed by comparing the correlations on the day before and after cold exposure.

We thank the reviewer for pointing out the diurnal pattern as a caveat. We now added metabolic cage data in the cold and at room temperatures spanning >236 hours. By quantifying the energy expenditure and food intake from

multiple light and dark phases we have excluded the possibility that the effects observed are simply due to measuring at different photoperiods (both remain increased, Extended Data Fig 1c-e).

The fiber photometry experiments (such as Figure 2F,J) would be more informative if the authors could show if the activity shown is specific to cold-induced feeding drive as opposed to a feeding drive due to an acute or chronic energy deficit. The studies of the CIED++ state raise this possibility by showing that an energy deficit before cold exposure enhances the responses.

We agree that providing specificity of Xi fiberphotometry in cold-induced feeding compared to feeding driven by fasting would strengthen our conclusion. As suggested, we performed fiberphotometry after overnight fasting and found that food deprivation-induced refeeding did not lead to the same increase in Xi activity (New Extended Data Fig. 5), indicating that Xi activity is specific for cold-induced feeding.

In the functional studies using chemogenetics to inhibit the previously activated Xi neurons, they should repeat the studies but without starving the animals beforehand. It is unnecessary and possible confounding to fast the animals as they should have elevated food intake during the period of cold exposure. They should also repeat the studies in fasted mice and in chronically malnourished mice at 30C, see <https://www.pnas.org/doi/10.1073/pnas.1002271107>

Fig. R1: Cold CAPTURED-Xi neurons with Gi or RFP were injected with CNO 4 hrs after being in cold with food and water, food intake was measured for next 1 hr. (same animals from Fig 3j)

As requested, we repeated the chemogenetic inhibition without the brief food restriction step (Fig. R1 here). We observed a similar decrease in food intake but more variability with the lowered baseline (p=0.056). Also, since now we have new calcium imaging data (new Extended Data Fig. 5) showing fasting-induced feeding did not lead to the increase of Xi activity, it is unlikely chemogenetic manipulation of Xi would affect fasting-induced feeding.

Inhibition of the Xi captured neurons decreases core temperature but it is unclear whether the reduced food intake leads to the reduced body temperature, as the authors suggest, or if the reduced body temperature decreases food intake. At a lower body temperature, energy expenditure is decreased thus diminishing the energy deficit. The authors should try to distinguish between these possibilities.

We thank the reviewer for pointing out this important caveat. Our new data (Fig. R2) indicated that inhibiting Xi at room temperature did not lead to a decrease in body temperature. Moreover, chemo-activation of Xi failed to alter body temperature in the cold or at room temperature. Thus, these data indicate that Xi neurons do not have a direct effect on body temperature. However, based on the feedback, we decided to remove the body temperature claims from the manuscript as it detracted from the main focus of the study.

Fig. R2: a-b, body temperature measurement after chemogenetic inhibition of captured Xi neurons (Gi), measured from implanted wireless probes 1 hour after CNO injection in the cold (a) or RT (b). (n=4 for RFP and n=8 for Gi). c-d, body temperature measurement after chemogenetic activation of captured Xi neurons (Gq) in the cold (c) or RT (d). (n=5 for RFP and Gq). Data are mean \pm SEM, ****P < 0.0001, ns, not significant, using unpaired t-test.

In the studies of RTPP, the authors should provide statistics comparing place preference at 4C to 30C.

We have included the statistics in the revised Fig 4I.

Overall the paper reports potentially interesting findings but further studies are required to confirm their conclusions.

Referee #2 (Remarks to the Author):

This study used a whole brain imaging method to identify key brain areas that are activated in response to cold-induced responses including increased energy expenditure and feeding. Based on a relatively long 4C cold exposure model (11 hr exposure ending on the onset of the dark cycle), it was identified that a few brain sites responded with c-Fos expression including a novel, functionally less known, mid thalamic region called the Xi nucleus. Using an established viral trapping approach, the Xi c-Fos neurons were permanently trapped and studied for its role during cold exposure and it was found that these neurons were active during cold-induced feeding, promoted feeding and transition from state 1 (energy conserving static) to state 2 (exploratory feeding). In addition, activation of these neurons was preferred only during 4C cold exposure.

The results on the Xi neurons in response to cold and promoting feeding behavior is completely novel and exciting, which should be interesting to broad readership. The experimental approaches are rigorous and the results are convincing with appropriate statistical analysis. The analysis of animal behaviors in 4C cold with 3 different states is also helpful in differentiating various physiologic states in cold exposure.

We appreciate that the reviewer found our work “completely novel and exciting”.

One major big-picture concern is that the results as presented seem to be a bit isolated in nature. What are the upstream and downstream neurons that may be important in mediating the observed effect? Given the large body of literature on various brain neurons that also respond to ambient temperatures and regulate metabolism, any relation of Xi neurons to those known neurons will help to build a big picture. Importantly, one previous study (Deem JD et al., Elife, 2020) reported a similar role for AgRP neurons in promoting feeding with cold exposure (14C). Is there any relationship between these 2 sets of neurons? Are AgRP neurons among those identified in the whole brain screening?

Thank you for this suggestion, we agree that integrating Xi with other established brain regions would increase the impact and broaden the horizon of our current manuscript. Based on this suggestion we have included a new figure Fig 5 and associated new Extended Data Fig. 11. We started off by looking for downstream regions of Xi and found that Xi projects to 3 main areas including Nucleus accumbens (NAc), Basolateral amygdala (BLA), and Anterior cingulate cortex (ACC) (Fig 5 a-b). This agrees with an earlier report (Ref 24). By using CTB retro-tracing coupled with cold-induced cFos double labeling, we found that cold-activated Xi neurons mostly project to NAc (Fig 5 c-g). Functional studies using projection-specific optogenetics showed that we can recapitulate the cold-capture Xi neurons phenotype for Xi-NAc projection but not Xi-ACC or -BLA projection (Fig 5h-l, Extended Data Fig. 11). Together these results helped us to knit in Xi with the NAc, a limbic region known to regulate feeding behaviors.

We have now included all the whole brain data as supplemental Table 1 which showed that there is an increase in cFos in the ARC. Based on the literature (Ref 9), these cFOS+ Arc neurons are likely AgRP+ neurons. However, we did not observe substantial Xi axons projecting to this area in our anterograde mapping experiments, suggesting that the ARC/AgPR population is not a direct downstream target of the Xi (thus outside the main scope of this paper).

One potential issue is the alignment of cold exposure with the diurnal pattern (Fig. 1a) used in this study. The 11hr 4C exposure was placed with an ending at the onset of the dark cycle, in which the increases in both energy expenditure and feeding during the onset of the ark cycle coincided with diurnal increases in both, leading to a concern that the observed increase in the late phase of cold exposure may not be due to a direct effect of cold exposure, thus the analysis on the relationship between increased energy expenditure and feeding presented in Fig. 1c may not be as the authors argued. It would be helpful to confirm this finding with the 11hr 4C exposure ending at the middle of the light cycle.

We thank the reviewer for pointing out the diurnal pattern as a caveat. We now added metabolic cage data in the cold and at room temperatures spanning 236 hours to include multiple day and night phases. Energy expenditure and food intake from multiple light and dark phases are quantified (both remain increased). Extended Data Fig 1c-f.

Data presented in Fig. 2a-2c with the 15min 4C exposure is somewhat confusing. This short cold exposure may too short to induce any changes including c-Fos expression or may be too late (5h later) to see c-Fos expression. Multiple durations of 4C exposure should be used with a shorter delay prior to tissue collection may be required to make the point.

We apologize for the misrepresentation of schematics in the original Fig 2a. The mice were exposed to 15 minutes of cold and perfused 90 minutes after that rather than depicted in the original schematics which gave the impression that mice were exposed to 15 minutes of cold followed by a 5 hour gap before perfusion. We have corrected the schematics in the revised Fig 2a. This 15min + 90min acute-cold paradigm was sufficient to induce cFOS expression in classic thermoregulatory regions like pre-optic area (New Extended Data Fig 4b) but failed to activate the Xi (Fig 2b), indicating only prolonged cold after CIEC led to Xi activation.

There appears to be inconsistency between data presented in Fig. 1 and Fig. 2. Fig. 2 showed that 5h 4C exposure activated a subset of Xi neurons, which led to increased feeding. However, Fig. 1 data showed that after 5h 4C exposure, the feeding was not increased until at a much later phase of the cold exposure. The authors need to clarify on this.

Thank you for pointing out this confusion. As you and other reviewers noted, the second increase of feeding in Fig 1b (around 10h) could be confounded by the diurnal pattern as the mice were entering the dark phase. As mentioned earlier, we now added metabolic cage data in the cold and at room temperatures spanning 236 hours to include multiple day and night phases. Energy expenditure and food intake from multiple light and dark phases are quantified and both were increased by the cold (Extended Fig. 1c-f). The 5-6h CIEC onset was defined based on the quantitative change of the correlation (rather than the increase in feeding alone) between EE and feeding which started between 5-6h and remain elevated afterwards (Fig 1c). We have updated line 72-78 to make this point clearer.

Fig. 3g data on Gq activation at 30C seem to be confusing. CNO-Gq induced activation of Xi neurons should mimic cold exposure and therefore activation of these neurons should promote feeding regardless of temperature exposed. However, the data presented showing Gq activation at 30C had no difference compared to the control group. Clarifications are needed for this point.

Thanks for pointing this out, we realized we did not convey the point clearly in schematics and in writing on the design of this experiment. The 30°C-Gq is a control group that has been injected with same AAVs but exposure to tamoxifen at 30°C. Since there is little cFOS expressed at the Xi region at 30°C, only few “background” neurons are expected to be captured at this temperature. Chemogenetic activation of these 30°C captured Xi neurons served as a negative control for the background labeling of the CAPTURE experiments. We now updated Fig 3g (as well as all other relevant figures) to highlight the “capture” vs. “testing” temperatures and updated the main text.

Referee #3 (Remarks to the Author):

The manuscript by Lal et al first performs a phenotypic analysis to determine the time course of stereotyped behaviors of mice during exposure to 4C. Next, they screen for brain Fos activity as a result of cold-induced food intake to identify the Xi as a region differentially activated by cold exposure compared to thermoneutrality. Using calcium monitoring and chemogenetic/optogenetic neural activity manipulations, they then determine that Xi neurons are activated immediately prior to feeding bouts in the cold and not warm, and that activity in these neurons bi-directionally influences cold-induced feeding. Overall, the findings are interesting and the experiments are scientifically sound. The manuscript is also well-written and overall an enjoyable read. The

weaknesses of the manuscript are that (1) some findings are preliminary and (2) the manuscript lacks mechanistic insight into how the Xi is involved in cold-induced feeding behavior. The following suggestions would improve the impact of the manuscript:

We thank the reviewer for the positive and constructive feedback.

Major comments:

1) Several comments regarding Figure 1:

- Quantitative and statistical comparisons between 23C and 4C data in Figure 1A-B should be included in the main figure, for both food intake and energy expenditure*

Thank you for this suggestion. We have included the quantification of Fig 1a-b in Extended Fig 1. a-b. In addition, we now added metabolic cage data in the cold and at room temperatures spanning 236 hours to include multiple day and night phases to take diurnal patterns into consideration. Energy expenditure and food intake from multiple light and dark phases are quantified (both increased). Extended Data Fig 1c-f.

- The HMM is a cool way to analyze behavior, but the cold-induced feeding behavior should be compared to behavior at 23C or 30C*

While HMM is effective in identifying hidden parameters within a dataset, it is less ideal to compare states across datasets. As an unbiased method, HMM will identify states based on the input sequences within the dataset, regardless of whether the baseline behavior sequences are the same or not across different datasets. For example, cold-related shivering and nesting behaviors will be completely absent at 30C, whereas HMM would still generate 3 new states with new behavior sequences. In other words, the states generated by unbiased HMM at one condition are not equivalent or comparable to those generated at a different condition.

- Since the cold is performed in the light period, why is sleep not included in the model?*

This would be an interesting feature to include. However, we do not think we have right hardware and techniques to precisely determine if the animal was asleep or just resting. Therefore, we assigned the immobile and resting postures in general as “Sit” in our manual annotation.

- Further, it is difficult to interpret the modeling data as depicted in Figure 1 and the related supplements. More descriptive explanations in text and/or legends would be very useful.*

We have elaborated on our description in the updated Figure 1 legend.

2) The unbiased screen indicated many regions, beyond the Xi, that are activated in 4C (compared to 30C). Is the Xi unique, compared to these other differentially activated regions, in its involvement in cold-induced food

intake? If neurons in these other regions are monitored and manipulated neurons in the other regions that were activated, do they also respond to or have an effect on cold-induced feeding?

We thank the review for suggesting this important clarification. We have now included screen results from all brain regions (Supplementary Table 1). We selected Xi for detailed characterization based on previous hamster lesion studies (Ref 22) and our follow-up temporal c-Fos quantification (Fig 2a-c); however, we do not propose Xi as the only region involved in cold-induced feeding and we now clarified this important point in the discussion (Line 288-289). It is not feasible to test each of the candidates from the whole-brain screen, but we now disclosed all the screen results for the community to freely explore. Moreover, rather than comparing Xi to other independent regions, we performed additional experiments to compare the regions where Xi neurons project to (New Fig 5) for their role in cold-induced feeding. We believe these data better demonstrated the specificity of Xi associated circuits.

3) The in vivo fiber photometry analyses are preliminary and raise a variety of further questions.

- Although the authors determined that Xi neurons are not responsive to temperature per se, responsivity was only measured during food intake in 4C and 30C. What happens at temperatures in between? Are there gradual increases in activity as temperature drops? Or is there a critical threshold? This kind of analysis, depending on results, could help the authors' speculations about the Xi as a behavioral gate.*
- An analysis of how $\Delta F/F$ relates to the size of the feeding bout may provide insight into the nature of the Xi activity. For example, since the calcium transient occurs prior to the bout, is it predictive of the amount of food intake?*
- Fiber photometry was performed only after 5 hours of cold exposure. Does Xi neuron activity in response to food intake change in the early cold vs the late (5+ h) cold? In other words, do Xi neurons respond to food only during the period of cold-induced feeding?*

We thank the reviewer for raising an important point about specificity of Xi neurons during cold-induced feeding and if there is any correlation between amount of food intake and calcium signals. (1) we performed fiberphotometry after overnight fasting at 30°C. We found that despite fasting-induced feeding, there was no activity peak observed before the feeding bouts as observed in cold-induced feeding (New Extended data Fig. 5). (2) we tested if intermediate temperature between 4°C and 30°C has any role. We performed Xi-fiberphotometry during the same fasting-refeeding experiment paradigm but this time mice were fasted and refed at 23°C. Again, we did not observe a calcium peak before the fasting-induced feeding bouts. Based on these results we concluded that Xi neurons only respond specifically to cold-induced feeding as opposed to a more general food response. Based on (1) and (2), together with the quantification of dF/F AUC (Extended data Fig. 5c), we think while the dF/F is scalable to CIEC, the scalability is specific to the cold but not the general amount of food intake in other (fasting) conditions.

Lastly, we did not observe activity increase as the temperature gradually dropped from 30°C to 4° either by simultaneous fiberphotometry recording (Extended Data Fig 4e) or by c-Fos staining (Fig 2a-c); while the preoptic area (MnPO, serving as the positive technical control) showed activation under the same conditions.

- Minor comment: line 122 indicates a correlation, but there are no correlations in the figure*

We modified this statement to be “Xi activity associated with” (Line 153) and throughout the manuscript.

• I appreciate that the authors included an example raw trace in the inserts for the averaged photometry traces.

We thank reviewer for acknowledging this!

4) Only about 50% of the vCAPTURE neurons activated by 4C display cold-induced Fos – a relatively low percentage. The authors should report overlap both ways – the % of Fos neurons that are captured, and the % of captured neurons that have Fos. This low overlap raises concerns about the vCAPTURE neuron specificity – the authors should at the very least discuss this or preferably address it experimentally by recording in vivo dynamics in vCAPTURE neurons.

We thank the reviewer for suggesting adding this important discussion (now included in the main text, line 170-174). Given the time frame of IEG expression (several hours followed by neuronal activity), all IEG-based labeling strategies (either via engineered promoters or endogenous staining) will inevitably capture a substantial number of neurons that were recruited by ongoing “baseline” activities before and after the stimuli. On the other hand, not all potentially activatable cells are recruited by a given stimulus, nor does the same repeated stimulus presentation mean the identical experience for the animal. The reported 50% overlap here is consistent with previously published vCAPTURE performance (Kingsbury, 2020; Kim & Ye, 2017), as well as other published IEG TRAP-like systems (Guenther 2013, Kawashima 2013, Ye 2016, Katsuyasu 2016). We also now included the suggestion % cFos/CAPTURE in the main results (Line 173), and listed all the raw cell counts in each animal below.

	30C-CAPTURE					4C-CAPTURE				
	Gq 30C	cFOS 4C	double+	Gq/Fos%	cFos/Gq%	Gq 4C	cFOS 4C	double+	Gq/Fos%	cFos/Gq%
	10.0	50.0	8.0	16.0	80.0	17.0	26.0	15.0	57.7	88.2
	13.0	37.0	6.0	16.2	46.2	21.0	29.0	21.0	72.4	100.0
	9.0	34.0	8.0	23.5	88.9	16.0	43.0	15.0	34.9	93.8
	7.0	16.0	3.0	18.8	42.9	18.0	34.0	17.0	50.0	94.4
	10.0	30.0	5.0	16.7	50.0	23.0	36.0	21.0	58.3	91.3
	11.0	43.0	6.0	14.0	54.5	24.0	35.0	21.0	60.0	87.5
	1.0	20.0	1.0	5.0	100.0	21.0	27.0	17.0	63.0	81.0
	11.0	43.0	9.0	20.9	81.8	23.0	34.0	21.0	61.8	91.3
	7.0	26.0	4.0	15.4	57.1	25.0	29.0	21.0	72.4	84.0
	2.0	13.0	1.0	7.7	50.0	11.0	21.0	9.0	42.9	81.8
	8.0	24.0	3.0	12.5	37.5	12.0	35.0	12.0	34.3	100.0
	10.0	36.0	7.0	19.4	70.0	9.0	21.0	9.0	42.9	100.0
	5.0	31.0	2.0	6.5	40.0	12.0	26.0	10.0	38.5	83.3
	9.0	26.0	3.0	11.5	33.3	18.0	31.0	15.0	48.4	83.3
	2.0	24.0	2.0	8.3	100.0	27.0	47.0	24.0	51.1	88.9
Mean	7.7	30.2	4.5	14.2	62.1	18.5	31.6	16.5	52.6	89.9
STD	3.7	10.4	2.7	5.5	22.8	5.6	7.3	4.9	12.4	6.7
SEM	0.9	2.7	0.7	1.4	5.9	1.4	1.9	1.3	3.2	1.7

5) For the neural activity manipulations, the experiments were always performed in the cold. What if instead these neurons were activated or inhibited at RT or thermoneutrality? Is food intake affected? These controls are important to determine whether these effects are specific to cold-induced food intake.

Thank you for suggesting these controls. We have performed the feeding experiments with optogenetics activation at RT or thermoneutrality (New Extended Data Fig.9 e and f) and found no change in the feeding. Similarly, chemogenetic activation/inhibition of these neurons did not affect food intake at RT (New Extended Data Fig 7b-c). We also established the specificity of Xi activity to cold-induced feeding using fiberphotometry during fasting refeeding experiments (see above answer to question#3, New Extended Fig 5).

6) Related, how specific are the Xi neuron responses to cold-induced food intake? Are they also activated by other kinds of actual or perceived energy deficits (e.g. exercise, glucoprivation?). Addressing these questions would provide more insight into the specificity of the observed effects.

We thank the reviewer for guiding us toward answering this very important question about specificity of Xi neurons during different energy deficit scenario. Based on your suggestion we performed Xi fiberphotometry for fasting-refeeding mice and aligned the calcium transits to feeding bouts. We found that unlike cold-induced feeding where we see a clear peak before feeding bout, during fasting-refeeding experiments we did not see calcium transits were aligned to feeding bout (New Extended Fig 5). Based on these results we concluded that Xi neurons play a specific role during cold-induced feeding. Please see answer to question #3 above for more detailed explanation.

7) In general, while the manuscript identifies Xi as a region mediating cold-induced food intake, there are no mechanistic underpinnings. The increase in energy expenditure precedes the food intake increases, how is this being signaled? Do these neurons receive inputs from BAT? Conversely, inhibiting these neurons prevents the cold-induced increases in temperature – do they connect to BAT via direct or polysynaptic connections?

We thank reviewer for pointing out this important limitation of our study. We have included a full new figure (Fig. 5) and its associated extended figure (Extended Data Fig. 11) exploring the downstream mechanism of action for Xi neurons. In short, our new data suggested Xi mediates feeding behavior through a projection to the NAc, a limbic reward center previously reported to regulate feeding (30,32). We also characterized canonical circulating factors known to be representing fasting, although they do not appear to be moving in the same direction in the cold (New Extended Data Fig 1f, Fig 6a-e). As for the BAT, considering the location of Xi, potential connectivity with the BAT will be polysynaptic/indirect and unlikely to be resolved within the scope of this manuscript.

8) The ns for each experiment widely vary. For experiments with very low n (e.g., 3-5), more subjects should be added.

We have increased the n for experiments where n was low (New Fig 4c-i, Extended Data Fig 9e-i)

9) The discussion section lacks depth and could be better used to interpret findings in the context of the literature.

We have expanded the discussion section.

Other comments:

1) *Lines 33-35 should include appropriate citations.*

We have included new citations.

2) *Figure 1 D/E, it is very difficult to distinguish between all of the colors.*

We have changed the color of the panel in the main figure.

3) *On line 150-151, the authors conclude that these neurons are necessary for survival in the cold, but this statement is quite premature as it is based on 1-h measurements of temperature.*

We have removed the statement in the text.

4) *Line 181, the authors should not report “slightly negative valence” as this is not statistically significant.*

We have corrected the statement in the text.

5) *Extended data Figure 4e,f – x axes labels are CEID rather than CEIC*

We have changed the label from CIED to CIEC.

Referee #4 (Remarks to the Author):

The work by Lal and collaborators assessed the role of the thalamic xiphoid neuronal activation as driver of cold-induced food intake. They showed that 5-6 hours of cold exposure induced neuronal activation in this midline thalamic structure by means of cFos and in vivo Ca²⁺ imaging. Using chemo- and optogenetics they revealed that the activation of neurons in this nucleus play a role in cold-induced feeding with an interesting positive valance. The role of this nucleus in regulating feeding is novel and the approach used is state-of-the-art.

However, there are some issues that need to be addressed, including the time points analyzed and the environmental temperatures used in several experiments. Furthermore, the identification of the phenotype of these neurons in the xiphoid nucleus and their projections fields requires further studies.

We thank the reviewer for detailed comments. We appreciate that the reviewer found our work novel and our approaches state-of-the-art.

Specific comments:

1) The authors assessed cold-induced feeding at 5h from the start of the exposure. However, from Fig. 1b a major increase in feeding can be observed from 2-3 hours from the start of the exposure. Thus, a time course on cold-induced cFos should be performed, including earlier and later time points to determine the correlation between the pattern of food intake and Xi activation.

We thank reviewer for highlighting importance about correlation of food intake and Xi activation.

First, based on the suggestion, we showed that at earlier time point (15 minutes cold + 90 min cFos induction, updated Fig 2a-c) at which classic thermoregulatory center MnPO has been activated (new Extended Data Fig 4b), Xi was not activated. At later time point (after 1 week of cold exposure, new Extended Data Fig 4c), Xi remained activated in both male and female mice. Together with Fig 2a-c, these data indicated prolonged cold is required to activate Xi. However, since cFos has limited temporal resolution (for example, once expressed cFOS protein could remain stable for several hours post-activation), it is not feasible to use cFOS alone to differentiate the finer timing differences with 2-5 hour window.

Next, to address the caveat regarding cFos, we used fiberphotometry to examine the dynamics of Xi neurons at finer temporal resolution. There was no Xi activity change when the ambient temperature was changed from 23 to 4°C, suggesting Xi was not activated by acute cold sensation (updated Extended Data Fig 4e).

Furthermore, to directly address the question about pattern of food intake and Xi activation, we designed two new experiments where we analyzed the food intake pattern during fasting-refeeding at room temperature or at thermoneutral with Xi fiberphotometry (New Extended Data Fig. 5a-b). When all fiberphotometry data were quantified (New Extended Data Fig 5c), we found that Xi activity is specific to cold-induced feeding but not fasting induced feeding at either room temperature or thermoneutrality.

2) I would also suggest for the authors to maintain consistency in the temperature used throughout the study. For example, the experiments in Fig. 1 were done switching either from 23°C to 4°C or from 30°C (thermoneutrality) to 4°C.

We agree that it would be ideal to have done the metabolic cage experiment in Fig 1 from 30°C (thermoneutrality) to 4°C. However, there are experimental limitations to the fast, new indirect calorimetry systems, including the Sable Systems Promethion used in this study. Changing temperatures from 30°C to 4°C within the experimental enclosure rapidly increases the relative humidity and wreaks havoc with the metabolic measurements. Our system can measure the metabolic rate of all cages every 2 minutes by taking an experimental shortcut. The energy expenditure is calculated from the VCO₂ and VO₂. CO₂ sensors are largely unaffected by humidity. However, the amount of oxygen in the air is highly dependent on humidity. To compensate, we record O₂, H₂O, and CO₂ levels from each cage. The O₂ levels are estimated based on steady-state water vapor readings. Rapidly changing humidity levels, as seen in a 30°C to 4°C transition, render the O₂ readings suspect until humidity levels reach equilibrium. To avoid these complications, we typically run our experiments from RT to 4°C.

Moreover, as new experiments at different temperatures have been requested as important controls by multiple reviewers, to improve clarity, we have now included additional labels throughout the figures to clearly indicate the temperatures at which each experiment was conducted.

3) For Fig. 1i, (the whole-brain SHIELD, extended data Figure 3), the authors should disclose all of their cFos studies throughout the brain as a resource for others.

We have added a new table disclosing the data about all the brain regions (Supplemental Table 1).

4) The authors need to determine whether 4°C-induced Xi neuronal activation can induce food intake at RT or 30°C. Moreover, some other areas including ARC and PVN should be investigated upon Xi-activation.

Thanks for suggesting these control experiments. With new experiments, we show that opto/chemogenetic activation/inhibition of cold captured Xi neurons did not affect food intake when activated in RT or 30°C (New Extended Data Fig 7b-c; Fig 9e-h). These results provide evidence that 4C-activated Xi neurons plays a specific role in cold-induced feeding. In addition, we also provided new fiberphotometry data of Xi during fasting-refeeding, showing that unlike cold-induced feeding where we saw a calcium peak before feeding bout, during fasting-refeeding we did not observe a peak before feeding bout (New Extended Data Fig 5c).

In new Fig 5 and Extended Data Fig 11, we found Xi projected to several brain regions including NAc, ACC, and BLA and identified the Xi->NAc projection as a key mediator for cold-induced feeding. In contrast, we did not observe significant Xi axonal terminals to ARC or PVN, suggesting their activity is unlikely to be, at least, directly affected by Xi activation.

5) Line 134, the authors mentioned “CIEC selectively induced cFos in the Xi without activating the surrounding areas (Fig. 2b),”, however, an increase in cFos expression can be seen in areas around Xi. These areas should be quantified as well.

We have added the quantification of surrounding Re area (New Extended data Fig 4a). We did not see a quantitatively significant change in cFOS expression in the neighboring region.

6) The synaptic activity-responsive element (SARE) is a neuronal activity-dependent enhancer that regulates the induction of the immediate-early gene (IEG) *Arc6*. Three activity-dependent transcription factors (CREB, MEF2 and SRF) bind to SARE and synergistically drive high expression of downstream genes. An enhanced version of the SARE promoter (E-SARE) drives downstream expression at much higher levels than previous IEG promoters. Thus, would the AAV-cFos-ER-Cre-ER a better viral approach for the experiments in Fig. 3 than the AAV-ESARE-ER-Cre-ER used in this study (since the authors determined cFos expression in Xi neurons)?

While we previously used AAV-cFos-ER-Cre-ER in Ai9 reporter mice (as the original CAPTURE) with great success (Ye et al, Cell, 2016), in subsequent work developing the viral CAPTURE (Kim & Ye et al, Cell, 2017) we found the stronger E-SARE was required to drive robust recombination when the payload was on another AAV vector. Others have also showed that E-SARE-based vCAPTURE maintained similar specificity to the endogenous IEGs (Kingsbury et al, Neuron, 2020).

7) In Fig. 3g, was food measured at 4°C, 30°C or RT? Does the activation of XiCIECneurons induce food intake at 30°C or RT? What about the food intake follow the activation of neurons “captured at 30°C” and tested at RT or 30°C? Similarly, what about food intake of hM4Di at RT?

Thanks for pointing this out as the original description of this experiment was not sufficiently clear. In Fig. 3g, food was measured at 4°C, we have updated the new Fig 3 to make it clear. The chemo-activation or inhibition of Xi CIEC neurons did not lead to changes in food intake without the cold (New Extended Data Fig 7b-c; Fig 9e-h). To improve clarity, we have now included additional subtitles/labels in all figures to clearly indicate the temperatures at which each experiment was conducted.

8) Similar to the chemogenetic experiments, in the optogenetic experiments of Fig. 4, how was the food intake and physical activity when Xi cold-captured neurons were activated at RT?

There was no change in food intake or physical activity when Xi cold-CAPTURED neurons were activated at RT or 30°C (New Extended Data Fig.9 e-h)

9) Regarding Extended data Fig6 (hSyn-ChR2), food intake and physical activity at RT need to be provided.

We have added the new Extended data Fig. 8f, which showed bulk Xi optogenetic activation did not increase food intake but led to increased physical activity.

Reviewer Reports on the First Revision:

Referees' comments:

Referee #1 (Remarks to the Author):

The authors set out to find the neurons responsible for cold-induced feeding by using a set of unbiased behavioral classification and activity mapping techniques. After finding that the xiphoid nucleus is activated by chronic (five hours) but not acute (15 minutes) of cold exposure, they analyzed the natural activity of these neurons using fiber photometry. This was followed by functional studies in which the effect of reactivation or reinhibition of Xi neurons previously activated in the cold (using the E-sare system) was tested using chemogenetics and optogenetics. The authors report that effects on food intake are specific to the cold temperature context and associated with positive valence. Consistent with this, studies of their projections identify the Nucleus accumbens as a downstream site that conveys the feeding and valence effects. In aggregate, the data are interesting and the identification of a neural substrate specific for cold-induced feeding is novel and of interest.

Overall the authors have done a good job responding to the prior critique. However the paper would be further improved if the authors addressed the following in a revised discussion:

- The authors should more fully discuss the possible mechanism responsible for activation of the xiphoid nucleus in the cold. Possibilities include hormonal or other interceptive inputs, direct sensing of temperature by these neurons or upstream neurons, sensing of metabolites that then cause plasticity of the neuronal inputs that are otherwise silent, or others.
- It is somewhat surprising that these neurons are not associated with positive valence even though they project to the NAcc. Could there be coincidence detection from another temperature sensing site that also projects to NAc?
- The authors note that feeding lags behind the temperature change using a scatterplot in Fig. 1c. To more fully address the possible effect of circadian rhythm it would be useful to plot feeding vs EE during the light phase the day before at room temperature i.e; 23C to show that there is a difference. The authors should already have these data.
- Since the authors do not use a genetically encoded marker specific for the Xiphoid, there is the possibility of transfecting cells in the neighboring areas. This is particularly important for their tracing experiments, considering that some areas (ACC and BLA) do not show much overlap in the subsequent CTB tracing. The authors should comment on some potential limitations of their viral targeting and potential effects of targeting neighbouring areas including the PVH.

Referee #2 (Remarks to the Author):

In this resubmission, the authors have clarified some of confusions and performed substantial additional experiments and the new results have adequately addressed the concerns raised during the initial review. The only remaining suggestion is that the authors should make it clearer the concept that the Xi neurons regulate feeding and valence in a cold-dependent manner, which means that there are other brain sites that sense cold and act in concert with the Xi neurons in promoting cold-induced feeding. In this sense, the authors' statement in the abstract "we found that optogenetic and chemogenetic stimulation of cold-activated Xi neurons recapitulated cold-induced feeding" is misleading. The dependence on the cold should be made explicit here. Also in this reviewer's view, this point should also be emphasized in the discussion.

Referee #3 (Remarks to the Author):

Overall, the authors addressed many of my and other reviewers' concerns to strengthen the data and conclusions presented in the manuscript. I have a few additional comments, and congratulate the authors on their exciting findings.

1. While the authors added data indicating that calcium dynamics in Xi neurons in response to food are much smaller at RT than the cold, they did not inhibit Xi neurons under acute or chronic food restriction to see if they attenuate increases in food intake – this remains an important control for conclusions on cold-specificity, as calcium activity is an excellent measure but does not necessarily provide a complete picture with regard to cellular activity.

2. It is puzzling to this reviewer that activation of Xi neurons trapped at 4C does not cause food intake (though, there appears to be a very small increase with optogenetic stimulation but not with chemogenetic stimulation). This result implies that it is not simply activity in these neurons that mediates cold-induced feeding, but rather there is an upstream thermoregulatory gate that is required, after which activity in Xi neurons is sufficient to drive feeding. This gate could be inputs from BAT, or other cold sensing or energy deficit sensing neurons, and as suggested in the first review, these upstream mediators will be very interesting and important to investigate. At the very least this should be included in interpretations. As written, the authors conclude that the Xi neurons may be a gate for feeding/behavioral switches, but taking all of the data into consideration, the gate (or, another gate) seems to be upstream.

Referee #4 (Remarks to the Author):

The authors have satisfactorily addressed all the points raised by this reviewer. One comment, which I did not pick up earlier, is that in figure 3g and j, the food intake of the control groups (4oC-RFP) are quite different, although these mice are in the exact same experimental conditions. Interestingly, the 4oC-RFP control group in 3g eats as much as the chemo-inhibited group in 3j while the control group in 3j eats as much as the chemo-excited group in 3g.

Author Rebuttals to First Revision:

Response to Referees' comments:

Referee #1 (Remarks to the Author):

The authors set out to find the neurons responsible for cold-induced feeding by using a set of unbiased behavioral classification and activity mapping techniques. After finding that the xiphoid nucleus is activated by chronic (five hours) but not acute (15 minutes) of cold exposure, they analyzed the natural activity of these neurons using fiber photometry. This was followed by functional studies in which the effect of reactivation or reinhibition of Xi neurons previously activated in the cold (using the E-sare system) was tested using chemogenetics and optogenetics.

The authors report that effects on food intake are specific to the cold temperature context and associated with positive valence. Consistent with this, studies of their projections identify the Nucleus accumbens as a downstream site that conveys the feeding and valence effects. In aggregate, the data are interesting and the identification of a neural substrate specific for cold-induced feeding is novel and of interest.

Overall the authors have done a good job responding to the prior critique. However the paper would be further improved if the authors addressed the following in a revised discussion:

We thank the reviewer for acknowledging that we did a “good job responding to prior critique”. In line with the reviewer’s suggestion, we have modified our discussion section to address their concerns.

- The authors should more fully discuss the possible mechanism responsible for activation of the xiphoid nucleus in the cold. Possibilities include hormonal or other interceptive inputs, direct sensing of temperature by these neurons or upstream neurons, sensing of metabolites that then cause plasticity of the neuronal inputs that are otherwise silent, or others.

We have added these points in our revised discussion.

- It is somewhat surprising that these neurons are not associated with positive valence even though they project to the NAcc. Could there be coincidence detection from another temperature sensing site that also projects to NAc?

This is an important point. We believe that the overall physiological state generated by prolonged cold exposure (a combination of thermo-sensing, and changes in metabolic and internal state) is required for Xi mediated CIEC effect on food intake. This has been added to the revised discussion as above.

- The authors note that feeding lags behind the temperature change using a scatterplot in Fig. 1c. To more fully address the possible effect of circadian rhythm it would be

useful to plot feeding vs EE during the light phase the day before at room temperature i.e; 23C to show that there is a difference. The authors should already have these data.

The cold exposure would inevitably disrupt the circadian rhythm on the experimental day, thus comparing 1c to another RT day would not be informative. Comparisons in Extended Data Fig 1c-e only included “stabilized” days/nights without the transition period can provide a better assessment of the potential effect of the circadian rhythm.

- Since the authors do not use a genetically encoded marker specific for the Xiphoid, there is the possibility of transfecting cells in the neighboring areas. This is particularly important for their tracing experiments, considering that some areas (ACC and BLA) do not show much overlap in the subsequent CTB tracing. The authors should comment on some potential limitations of their viral targeting and potential effects of targeting neighbouring areas including the PVH.

We have revised our discussion (Line 229-231) to include this point.

Referee #2 (Remarks to the Author):

In this resubmission, the authors have clarified some of confusions and performed substantial additional experiments and the new results have adequately addressed the concerns raised during the initial review. The only remaining suggestion is that the authors should make it clearer the concept that the Xi neurons regulate feeding and valence in a cold-dependent manner, which means that there are other brain sites that sense cold and act in concert with the Xi neurons in promoting cold-induced feeding. In this sense, the authors' statement in the abstract "we found that optogenetic and chemogenetic stimulation of cold-activated Xi neurons recapitulated cold-induced feeding" is misleading. The dependence on the cold should be made explicit here. Also in this reviewer's view, this point should also be emphasized in the discussion.

We appreciate that the reviewer thinks we have “adequately addressed the concern raised during initial review”. We have now expanded our discussion to more fully capture the essence cold dependence during feeding by Xi neurons. We modified our abstract to: “*we found that optogenetic and chemogenetic stimulation of cold-activated Xi neurons selectively recapitulated food-seeking behaviors in the cold*” to better represent the finding.

Referee #3 (Remarks to the Author):

Overall, the authors addressed many of my and other reviewers' concerns to strengthen the data and conclusions presented in the manuscript. I have a few additional comments, and congratulate the authors on their exciting findings.

We thank the reviewer for all the constructive suggestions throughout the review process.

1. While the authors added data indicating that calcium dynamics in Xi neurons in response to food are much smaller at RT than the cold, they did not inhibit Xi neurons under acute or chronic food restriction to see if they attenuate increases in food intake – this remains an important control for conclusions on cold-specificity, as calcium activity is an excellent measure but does not necessarily provide a complete picture with regard to cellular activity.

Since Xi neurons do not show endogenous calcium dynamics with RT fasting/refeeding, experimentally inhibiting them below the baseline level could potentially introduce artifacts that can be hard to interpret regardless of the outcome. We would respectfully argue that it would be beyond the scope of the current manuscript.

2. It is puzzling to this reviewer that activation of Xi neurons trapped at 4C does not cause food intake (though, there appears to be a very small increase with optogenetic stimulation but not with chemogenetic stimulation). This result implies that it is not simply activity in these neurons that mediates cold-induced feeding, but rather there is an upstream thermoregulatory gate that is required, after which activity in Xi neurons is sufficient to drive feeding. This gate could be inputs from BAT, or other cold sensing or energy deficit sensing neurons, and as suggested in the first review, these upstream mediators will be very interesting and important to investigate. At the very least this should be included in interpretations. As written, the authors conclude that the Xi neurons may be a gate for feeding/behavioral switches, but taking all of the data into consideration, the gate (or, another gate) seems to be upstream.

We absolutely agree that additional cold-specific inputs are upstream to the Xi gating. We now expanded our discussion (2nd Paragraph in the Discussion) to include potential upstream mediators and highlighted their importance for future investigations.

Referee #4 (Remarks to the Author):

The authors have satisfactorily addressed all the points raised by this reviewer. One comment, which I did not pick up earlier, is that in figure 3g and j, the food intake of the control groups (4oC-RFP) are quite different, although these mice are in the exact same experimental conditions. Interestingly, the 4oC-RFP control group in 3g eats as much as the chemo-inhibited group in 3j while the control group in 3j eats as much as the chemo-excited group in 3g.

We thank the reviewer for all the inputs and are delighted that we have “satisfactorily addressed all the points raised by this reviewer”. The baseline difference in Fig 3g and 3j is because 3g is the normal CIEC condition but in 3j we created a CIEC++ situation by briefly restricting food in the cold to create a hyper-energy deficit state. Hence the 3j has higher baseline food intake.